EMBO
reports

# The ER-SURF pathway uses ER-mitochondria contact sites for protein targeting to mitochondria

Christian Koch[1], Svenja Lenhard[1], Markus Räschle [ID][2], Cristina Prescianotto-Baschong[3], Anne Spang [ID][3] & Johannes M Herrmann [ID][1][✉]

## Abstract

**Most mitochondrial proteins are synthesized on cytosolic ribosomes and imported into mitochondria in a post-translational reaction. Mitochondrial precursor proteins which use the ER-SURF pathway employ the surface of the endoplasmic reticulum (ER) as an important sorting platform. How they reach the mitochondrial import machinery from the ER is not known. Here we show that mitochondrial contact sites play a crucial role in the ER-to-mitochondria transfer of precursor proteins. The ER mitochondria encounter structure (ERMES) and Tom70, together with Djp1 and Lam6, are part of two parallel and partially redundant ER-to-mitochondria delivery routes. When ER-to-mitochondria transfer is prevented by loss of these two contact sites, many precursors of mitochondrial inner membrane proteins are left stranded on the ER membrane, resulting in mitochondrial dysfunction. Our observations support an active role of the ER in mitochondrial protein biogenesis.**

**Keywords** Contact sites; Endoplasmic reticulum; ERMES; Mitochondria; Protein import
**Subject Categories** Membranes & Trafficking; Organelles

## Introduction

Mitochondria consist of about 900 (yeast) to 1300 (human) different proteins (Morgenstern et al, 2017; Rath et al, 2021). Except for a handful of mitochondrially encoded proteins, all these proteins are encoded by nuclear genes and synthesized on cytosolic ribosomes. About two-thirds of these mitochondrial precursor proteins carry amino-terminal presequences (Vögtle et al, 2009), which serve as matrix-targeting signals (MTSs). The remaining mitochondrial proteins have internal targeting signals, which are commonly found in proteins of the outer membrane, the intermembrane space (IMS), and the inner membrane of mitochondria (Chacinska et al, 2009). Moreover, internal MTSs (iMTSs) are also found in many presequence-containing matrix

proteins since they facilitate their interaction with the mitochondrial outer membrane protein Tom70 (Backes et al, 2021; Backes et al, 2018). Tom70 and other receptors recognize the targeting signals and direct precursors into the protein-translocation pore of the translocase of the outer membrane of mitochondria, the TOM complex (Araiso et al, 2019; Ramage et al, 1993). Additional translocases in the outer (SAM complex, sorting and assembly machinery of the outer membrane) and the inner membrane (TIM22 and TIM23 complexes) sort precursors to their respective mitochondrial subcompartment. Owing to a very powerful in vitro import system based on isolated yeast mitochondria and radiolabeled precursor proteins, the different steps of the mitochondrial import reaction were elucidated in great detail. The mechanistic features by which the different translocases recognize their substrates and facilitate precursor translocation across membranes are, therefore, relatively well understood (Busch et al, 2023). Studies in mammalian cells revealed a high degree of conservation among eukaryotes, and the mitochondrial translocases of yeast and animal cells are of similar structure and operate by the same principles (Palmer et al, 2021).

In contrast, early steps in mitochondrial protein targeting are much less understood (Avendano-Monsalve et al, 2020; Bykov et al, 2020). Only a small subset of predominantly hydrophobic inner membrane proteins reaches the mitochondrial surface co-translationally (Williams et al, 2014). Therefore, initially, most precursors transiently explore the cytosol and interact with cytosolic chaperones (Becker et al, 1996; Deshaies et al, 1988; Drwesh et al, 2022; Krämer et al, 2023). Presequences are bound by cytosolic chaperones as soon as they emerge from the exit tunnel of the cytosolic ribosome and 80% of all mitochondrial precursors were found as substrates of the Ssb-type Hsp70 proteins (Doring et al, 2017).

Mitochondrial precursors can also associate with the surface of the endoplasmic reticulum (ER). ER-binding of precursors occurs via specific recruitment by the signal recognition particle (SRP) or by components of the guided entry of tail-anchored proteins (GET) complex as well as by unspecific association (Chartron et al, 2016; Costa et al, 2018; Vitali et al, 2018; Xiao et al, 2021). The P5A-ATPase (Spf1 in yeast, ATP13A1 in humans) recognizes and removes a subset of such ER-stranded mitochondrial precursors to facilitate their productive translocation to mitochondria (McKenna et al, 2022; McKenna et al, 2020; Qin et al, 2020).

[1]Cell Biology, University of Kaiserslautern, Kaiserslautern, Germany. [2]Molecular Genetics, University of Kaiserslautern, Kaiserslautern, Germany. [3]Biozentrum, University of Basel, 4056 Basel, Switzerland. [✉]E-mail: Hannes.herrmann@biologie.uni-kl.de

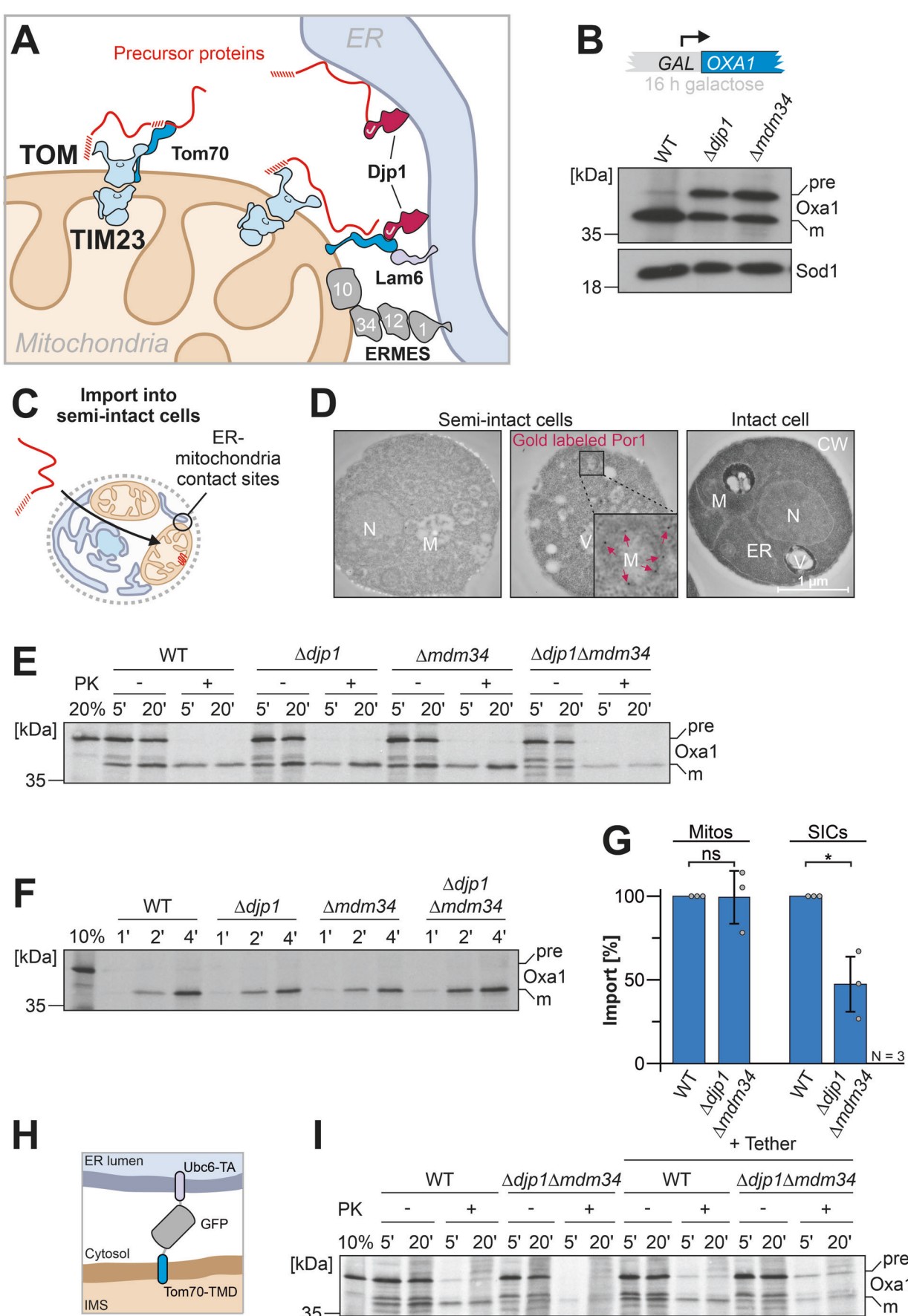

◄ **Figure 1.  Djp1 and ERMES cooperate in ER-SURF targeting.**

(A) Schematic representation of the ER-SURF pathway. Precursor proteins can be directly bound from the cytosol to mitochondria or via the ER surface. The ER-bound Djp1 supports precursor targeting via the ER. Djp1 and the ER membrane protein Lam6 form contacts with Tom70. The ERMES components form another, independent ER-mitochondria contact site. (B) The ER-SURF substrate Oxa1 was expressed from a strong *GAL* promoter. Cells of the indicated strains were shifted from lactate medium to lactate medium that contains 0.5% galactose for 4 h. Whole cell extracts were analyzed by Western blotting. The precursor (pre) and mature (m) species of Oxa1 are indicated. (C) Schematic representation of an import reaction into semi-intact yeast cells (SICs). SICs maintain the intracellular architecture and maintain organellar structures and contacts. (D) Wild-type cells were converted to semi-intact cells (see Materials and Methods) or directly used for electron microscopy. In the central image, the mitochondrial outer membrane protein Por1 was labeled by immunogold staining and highlighted by red arrows. (E) Radiolabeled Oxa1 was synthesized in reticulocyte lysate in the presence of $^{35}$S methionine and incubated with semi-intact cells derived from the indicated strains. After 5 or 20 min, the cells were isolated, treated without or with proteinase K (PK) for 30 min on ice, and subjected to SDS-PAGE and autoradiography. 20% of the radioactive Oxa1 protein used per import reaction (time point) was loaded for comparison. (F) Radiolabeled Oxa1 was imported into isolated mitochondria and treated as described for (E). (G) Quantification of import experiments with semi-intact cells and mitochondria, respectively. Shown are the mean values and standard deviations of three biological replicates. Statistical difference was calculated with a student's *t*-test. ns non significant, **p* value <0.05. (H) Schematic overview of the Chimera tether (Kornmann et al, 2009). The transmembrane domain of Tom70 (residues 1–30) was fused to GFP and to the c-terminal tail anchor sequence of Ubc6 (residues 233–250). (I) Indicated cells expressing Chimera or carrying an empty vector were converted to semi-intact cells and used for an import experiment equivalent to that described for (E). Please note that the expression of the tether did not suppress the import defect in the double mutant. Source data are available online for this figure.

We recently discovered that the binding of mitochondrial precursor proteins to the ER surface is not simply the result of an unavoidable mislocalization, but rather that the ER-to-mitochondria targeting of proteins via the ER-SURF route can increase the productive biogenesis of mitochondrial precursors (Hansen et al, 2018; Koch et al, 2021). The ER-bound J domain protein Djp1 increases the efficiency of ER-SURF targeting (Hansen et al, 2018; Koch et al, 2021; Papic et al, 2013). How precursor proteins are transferred from the ER surface to mitochondria is not known. The J domain of Djp1 might stimulate the transfer to cytosolic Hsp70 chaperones and help to release ER-bound precursors back into the soluble fraction. Alternatively, contact sites between the ER and mitochondria might mediate a direct transfer (Fig. 1A). A recent study showed that Djp1 binds to Tom70, and thus, Djp1 might directly contact mitochondria (Opalinski et al, 2018). Tom70 also makes contact with the ER protein Lam6, and its role in ER-mitochondria contact sites is established both in yeast and in humans (Elbaz-Alon et al, 2015; Filadi et al, 2018; Murley et al, 2015). The best-characterized ER-mitochondria contact site is formed by local clusters of about 25 copies of the ER-mitochondria encounter structure, ERMES (Kawano et al, 2018; Kornmann et al, 2009; Wozny et al, 2023). The chain-like Mdm10-Mm34-Mdm12-Mmm1 tetramer of ERMES mediates the passage of lipids via dedicated binding sites within Mdm34, Mdm12, and Mmm1 (Jeong et al, 2017; Jeong et al, 2016; Kawano et al, 2018; Kopec et al, 2010), but no direct role for ERMES in protein targeting has been established.

Single deletions of either Tom70 or ERMES are rather well tolerated; however, double deletions are inviable on all carbon sources (Backes et al, 2021; Murley et al, 2015). Thus, the presence of at least one of the two established ER-mitochondria contact sites is essential for viability in yeast, illuminating the important roles of ER-mitochondria contacts in maintaining organellar identity and/or function.

Here we describe that ER-mitochondria contacts mediated by ERMES and Tom70 are crucial for ER-SURF targeting. Simultaneous depletion of both types of contact sites strongly impairs the transfer of proteins from the ER to mitochondria and leads to selective reduction of mitochondrial proteins. Particularly proteins of the inner mitochondrial membrane are affected under these conditions, consistent with the observations that hydrophobic inner membrane proteins predominantly embark on the ER-SURF targeting pathway.

## Results

### Djp1 and Mdm34 cooperate in protein biogenesis

We previously identified Djp1 as a component that promotes the targeting of mitochondrial precursor proteins via the ER surface (Hansen et al, 2018). The relevance of ER-mitochondria contact sites for the ER-SURF targeting route has not been determined yet. We therefore expressed the precursor form of the mitochondrial inner membrane protein Oxa1, an established ER-SURF substrate, from a galactose-inducible *GAL* promoter, which results in a moderate overexpression of the protein. As previously reported, this leads to an accumulation of the precursor form of Oxa1 in *Δdjp1* cells, indicative of a compromised import of the Oxa1 precursor into mitochondria (Hansen et al, 2018). We found that a comparable accumulation of Oxa1 precursor was observed in cells lacking the ERMES protein Mdm34 (Fig. 1B), even though the levels of Djp1 protein were not affected in this mutant (Fig. EV1A,B). Moreover, when a fusion protein of Oxa1 and Ura3 was expressed in the *Δmdm34* mutant, cells showed an increased growth on uracil-deficient plates, indicative of the cytosolic accumulation of the Oxa1-Ura3 fusion protein (Fig. EV1C,D). Thus, the deletion of the ERMES subunit Mdm34 apparently reduces the import efficiency of the ER-SURF substrate Oxa1.

Next, we employed an in vitro import assay with semi-intact yeast cells (Beckers et al, 1987; Laborenz et al, 2021). For this, cells were converted to spheroplasts by removing the cell wall with zymolyase before the plasma membrane was partially permeabilized to allow the access of macromolecules to organelles that still maintain their physiological contacts and shapes (Fig. 1C). By electron microscopy, we verified the unperturbed intracellular organization of these spheroplasts including that of mitochondria which we marked by immunogold labeling of the outer membrane protein Por1 (Fig. 1D). We then synthesized radiolabeled Oxa1 precursor in reticulocyte lysate and incubated it with the semi-intact cells as described previously (Laborenz et al, 2021). The Oxa1

precursor was efficiently imported into semi-intact cells isolated from wild type (WT) or from mutants lacking either Djp1 or the ERMES protein Mdm34 (Fig. 1E,G). However, when both Djp1 and Mdm34 were absent, only a small amount of mature, protease-resistant Oxa1 was generated (Fig. 1G). Thus, the presence of either Djp1 or ERMES was critical for efficient mitochondrial import of Oxa1 in semi-intact cells. This import defect was not seen when the Oxa1 precursor was added to mitochondria isolated from Δdjp1 Δmdm34 cells indicating that the mitochondria of the double mutant per se are still fully import-competent (Fig. 1F,G).

Using semi-intact cells for an import reaction, a dependence on Djp1 and ERMES was also observed for Coq2, another substrate of the ER-SURF pathway (Hansen et al, 2018), but not for Hsp60 or Mrpl15, whose import occurs independent of the ER surface (Fig. EV1E–G). This suggests that ERMES and Djp1 are part of two distinct parallel-acting delivery routes, which can hand mitochondrial precursor proteins over from the ER surface to mitochondria. We assume that they play an additional role besides providing proximity through tethering of the two organelles as the artificial Tom70-GFP-Ubc6 tether construct, which bridges mitochondria and ER membranes (Kornmann et al, 2009), did not mitigate the import defect observed in the semi-intact cell assay (Fig. 1H,I).

## An Mdm34 depletion model to study the function of ERMES in protein biogenesis

The deletion of ERMES subunits such as Mdm34 leads initially to a severe growth phenotype on non-fermentable carbon sources (Kornmann et al, 2009); however, after a few generations, mutant cells adapt and the mutant phenotype is suppressed (Fig. EV2A,B). We, therefore, developed a more reliable CRISPR interference-based depletion system which made use of the expression of an *MDM34*-specific guide RNA under the control of a Tet repressor (Backes et al, 2021; Smith et al, 2016). The addition of anhydrotetracycline (ATc) induced the binding of a fusion protein of dCas9 and the transcription repressor Mxi1 to the *MDM34* promoter (Fig. 2A) and efficient repression of *MDM34* transcription (Fig. 2B), as well as of the Mdm34 protein (Fig. 2C,D). The depletion of Mdm34 resulted in the formation of hyper-fused, condensed mitochondria (Fig. 2E) on galactose-containing media, that are a hallmark of ERMES mutants (Dimmer et al, 2002; Kakimoto et al, 2018; Kornmann et al, 2009). Moreover, the depletion of Mdm34 caused a considerable growth defect of respiring cells and reduced growth rates in galactose and glycerol, respectively (Fig. 2F). The concentrations of ATc used (960 ng/ml) did neither inhibit mitochondrial translation nor cell growth (Fig. EV2C,D). In summary, the CRISPR interference-mediated knock-down of Mdm34 allows the controlled depletion of ERMES contact sites from yeast cells without giving cells a chance to adapt to the ERMES deficiency.

## Mdm34 and Tom70 are central components of parallel ER-SURF routes

The simultaneous deletion of the genes for Mdm34 and Tom70 is not possible as double null mutants are not viable (Backes et al, 2021; Murley et al, 2015). This synthetic lethal phenotype is presumably explained by the overlapping functions of ERMES and Tom70 in their role as tethers of ER and mitochondria (Elbaz-Alon

et al, 2015; Kornmann et al, 2009; Murley et al, 2015). We, therefore, employed the knock-down strategy for Mdm34 to generate cells in which Tom70 and Mdm34 were absent and depleted, respectively, and which therefore lack the established ER-mitochondria contacts (Fig. 3A). Growth assays on plates confirmed the strong synthetic growth defect in the Δtom70 background, particularly on the non-fermentable carbon source glycerol (Fig. 3B). *MDM34* repression in Δdjp1 or Δlam6 backgrounds also showed negative genetic interactions, albeit to a lesser extent, consistent with their redundant role as ER-bound Tom70 interactors (Fig. 3A) (Elbaz-Alon et al, 2015; Hansen et al, 2018; Murley et al, 2015; Papic et al, 2013).

Having established a strategy to deplete both ER-mitochondria contact sites, we next measured the consequences on the cellular proteome. To this end, we knocked down *MDM34* in wild type and Δtom70 cells for 0, 8, and 24 h and measured cellular proteomes of three independent biological replicates (Fig. 3C). Principle component analysis showed that the depletion of Mdm34 left a characteristic footprint on the cellular proteome (Fig. 3D) which was largely caused by effects on the mitochondrial proteome as apparent from an unbiased GO term analysis (Fig. EV3A). Depletion of Mdm34 resulted in a strong reduction of many mitochondrial proteins in the presence of Tom70 (Fig. 3E). This effect was more pronounced in the absence of Tom70 (Fig. EV3B). In particular, the levels of proteins of the matrix and, to an even higher degree, those of the inner membrane were significantly depleted in the Mdm34 knock-down cells, whereas most proteins of the outer membrane and the IMS were not affected (Figs. 3F and EV3C). Thus, ER-mitochondria contact sites are crucial for the biogenesis or stability of many mitochondrial proteins, especially those of the matrix and the inner membrane.

## ER-mitochondria contacts are crucial for the molecular identity of mitochondrial membranes

To elucidate the relevance of ER-mitochondria contact sites for the mitochondrial protein compositions, we isolated mitochondria from cells containing or lacking Mdm34 and Tom70 (Fig. 4A) using an established fractionation procedure on sucrose-based step gradients (Taskin et al, 2023). The "mitochondrial fraction" obtained from wild-type cells contained, as expected, predominantly mitochondrial proteins, whereas only a low amount of peptides were detected from non-mitochondrial proteins, indicative of highly purified mitochondria, consistent with previous reports (Morgenstern et al, 2017; Sickmann et al, 2003). However, in the absence of ER-mitochondria contact sites, many proteins of other cellular compartments were found in these fractions, in particular proteins of the ER/nucleus and of peroxisomes (Fig. 4B). The considerable accumulation of non-mitochondrial proteins could be due to defects in the accuracy of the intracellular protein targeting in these mutants or, though not mutually exclusive, to differences in the physicochemical properties of the different organelles which might affect their separation on centrifugation gradients.

We therefore analyzed the structure of mitochondria and other cellular compartments using transmission electron microscopy (Laborenz et al, 2019) and light microscopy with fluorescent ER- and mitochondria-targeted proteins. The size and overall volume of mitochondria in wild type and Δtom70 were comparable, but depletion of Mdm34 resulted in fewer but much larger mitochondria (Figs. 4C,D and EV4A). This nicely recapitulates previous

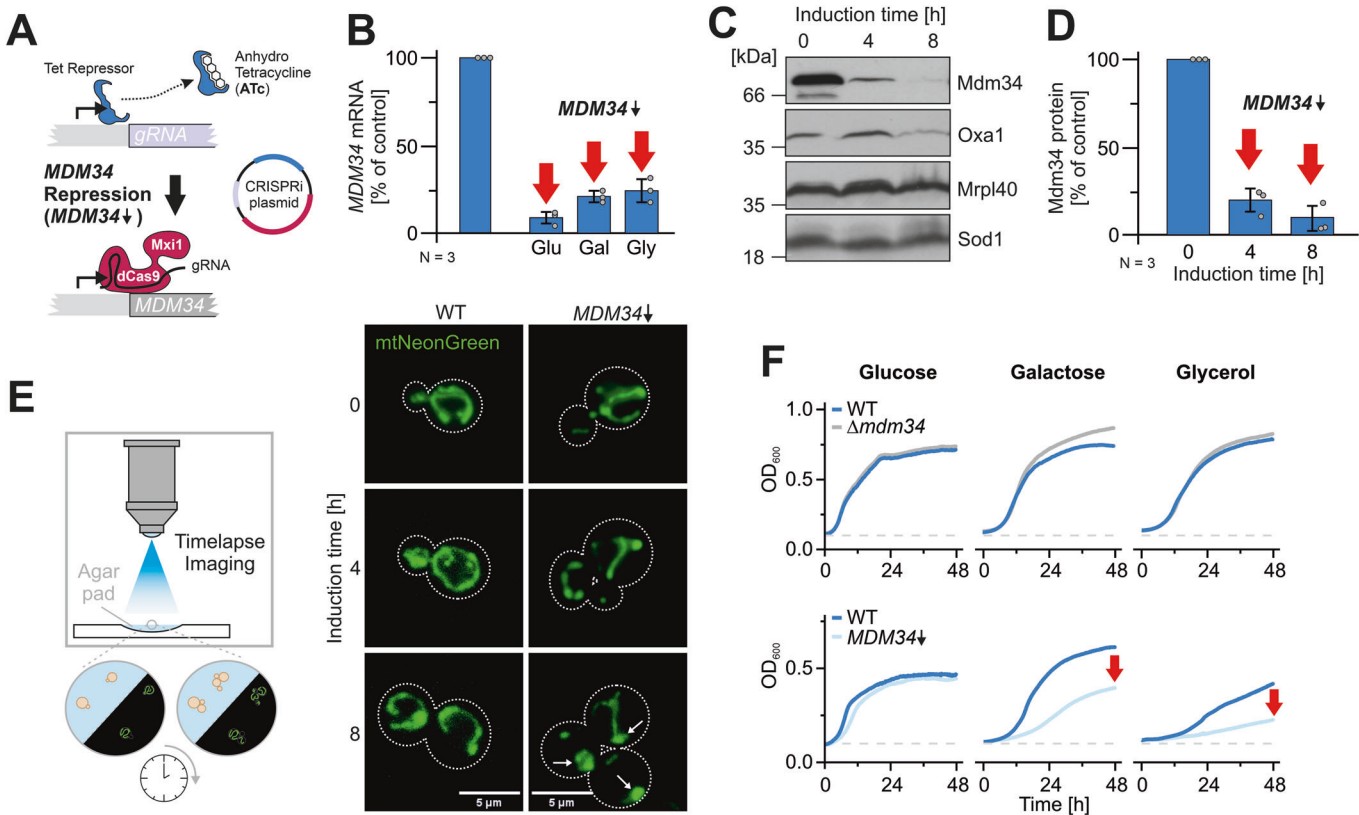

**Figure 2. A Mdm34 depletion model to study the function of ERMES in protein biogenesis.**

(A) Schematic representation of the *MDM34* depletion by CRISPR interference (CRISPRi) (Smith et al, 2016). ATc-induced inhibition of the Tet repressor leads to the expression of a gRNA that recruits the dCas9-Mxi1 fusion to the *MDM34* promoter, thereby blocking transcription of the *MDM34* gene. (B) A Mdm34-CRISPRi plasmid coding for the gRNA under Tet repressor control as well as for the dCas9-Mxi1 fusion, was transformed into wild-type cells. Cells were grown to the early log phase in a medium containing the indicated carbon sources (2% each) before 960 ng/ml ATc was added. The mRNA levels of *MDM34* as well as control transcripts, were quantified by qPCR before and 4 h after ATc-induced repression. Shown are the mean values and standard deviations of three biological replicates. Since the depletion efficiency depends on gene repression and dilution of existing transcripts, the depletion was quicker on glucose, where cells grow more rapidly than on glycerol. (C, D) *MDM34* expression was repressed in glucose-grown cultures of cells harboring a chromosomally tagged *MDM34* allele for expression of an Mdm34 protein with a C-terminal hemagglutinin (HA) tag. The protein levels of Mdm34-HA and of control proteins were analyzed by Western blot and quantified. Panel **D** shows the mean values and standard deviations of three biological replicates. (E) Wild-type cells were transformed with the Mdm34-CRISPRi plasmid (or an empty plasmid control) and with a plasmid for the expression of a mitochondria-targeted NeonGreen protein (Lenhard et al, 2023). The cells were grown on agar pads that contain galactose as a carbon source to better visualize the mitochondrial network. The mitochondrial network was continuously analyzed in a wide field microscope using a Leica 100x objective in a Dmi8 Thunder Imager. Whereas the mitochondrial network was maintained in the control cells, the network collapsed in *MDM34*-depleted cells, similar to the spherical mitochondria described before for Δ*mdm34* strains (Dimmer et al, 2002). (F) The indicated strains were grown in galactose medium to log phase and used to inoculate cultures with the carbon sources indicated. Cells were grown at 30 °C under constant agitation. Cell growth was continuously monitored. The graphs show the mean values of three technical replicates. Source data are available online for this figure.

observations about the relevance of ERMES for mitochondrial division (Boldogh et al, 2003; Dimmer et al, 2002; Ellenrieder et al, 2016; Sogo and Yaffe, 1994). Changes in the overall morphology of most of the other organelles were not observed; however, the vacuoles in Mdm34-depleted cells were often swollen and filled with proteins and probably also lipids (Fig. 4C, structures indicated with "V"). Despite the considerably changed morphology of mitochondria, we did not observe effects on the major cellular stress response pathways such as the heat shock response, the proteasome-associated control response, the pleiotropic drug resistance response (Fig. 4E), or the unfolded protein response (Fig. EV4B). Thus, the absence of ER-mitochondria contact sites changes mitochondrial properties, which perturbs the isolation of ultrapure mitochondria from fractionated cells via centrifugation gradients.

## Mitochondria and ER membranes can be purified by affinity purification

We next developed a strategy to purify mitochondria and ER from yeast cells by an alternative strategy using an affinity purification strategy based on recently published procedures (Reinhard et al, 2023). To this end, we expressed fusion proteins in the outer mitochondrial (Tom20) and the ER membrane (Sec63 and Rtn1) with a tandem affinity tag consisting of myc and flag epitopes separated by a protease 3C cleavage site (Fig. 5A). We chose Tom20 and Sec63 for purification since translocation complexes serve as entry gates into these organelles thereby defining the molecular identity of organelles. Mass spectrometry of the isolated fractions verified efficient affinity purification of the tagged proteins and the strong enrichment of ER proteins with Sec63 and Rtn1, and of

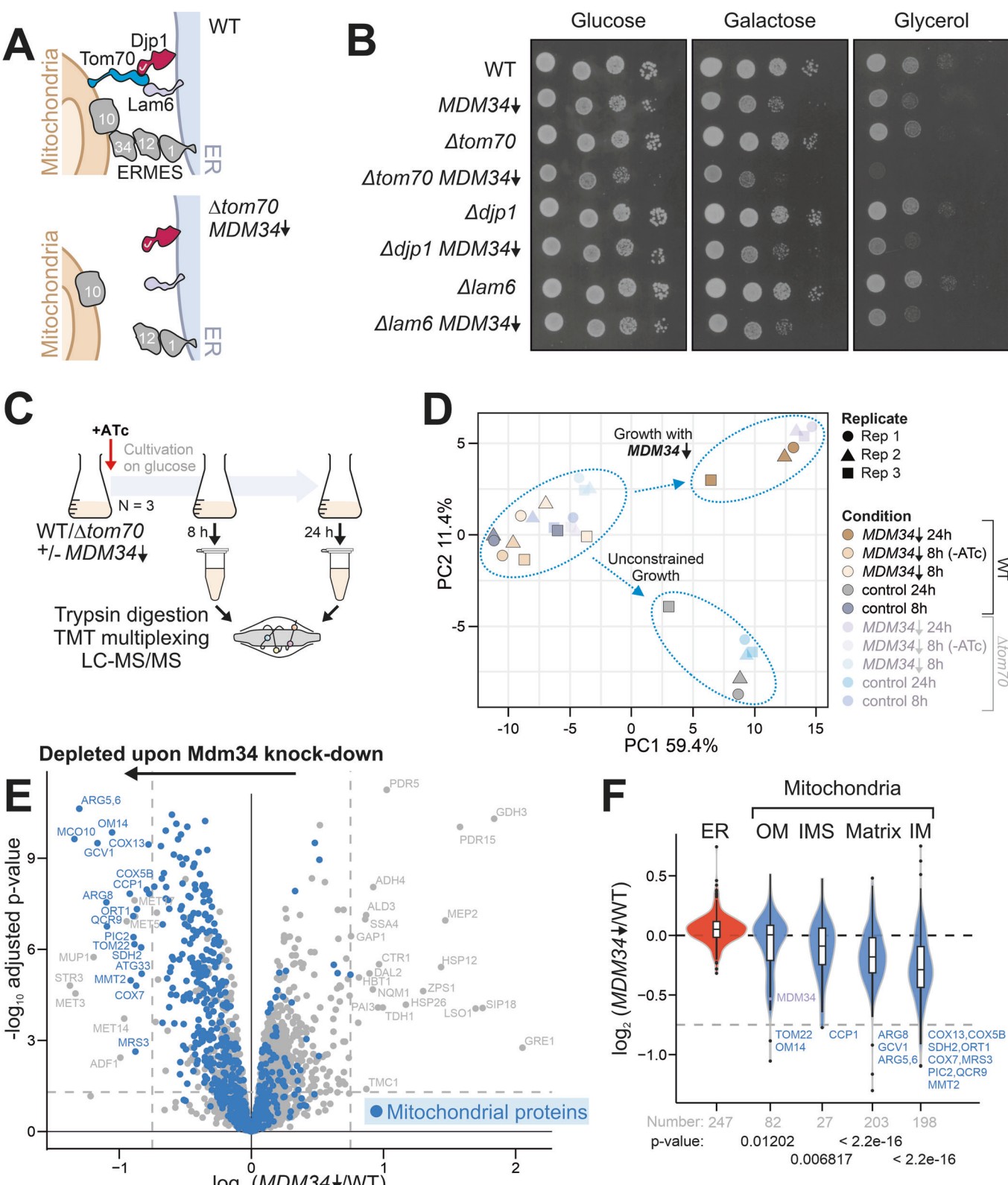

◄ **Figure 3. Depletion of ER-mitochondria contact sites leads to a specific reduction of inner membrane proteins.**

(A) The ERMES complex and Tom70 connect the ER to mitochondria. Djp1 and Lam6 serve as ER-bound interactors of Tom70. Cells were grown in glucose-containing media and grown for 8 and 24 h in the presence of ATc. Afterward, samples were taken, cells were lysed, and the resulting lysate was analyzed via LC-MS/MS. (B) Cells of the indicated strains were grown to log phase in galactose medium before tenfold serial dilutions were dropped onto plates with the indicated carbon sources. (C) Scheme of the proteome analysis of cells before and after Mdm34-depletion in wild type and Δtom70 cells. (D) Principal component analysis of the data set. Please note, that depletion of Mdm34 induces a characteristic and consistent protein pattern of the cellular proteome. (E) Comparison of the proteomes of wild-type control and Mdm34-depleted cells 24 h after ATc addition. Mitochondrial proteins (Morgenstern et al, 2017) were indicated in blue. For the calculation of fold changes and p values, the limma package within the R programming language was used (Ritchie et al, 2015). (F) The violin plot shows the ratio of protein abundance (log2-fold enrichment scores) in control relative to Mdm34-depleted cells. Numbers below the different subclasses represent the number of proteins (n) within a given subclass. Boxes represent the data range from the first (Q1) to the third quartile (Q3), with the line in the middle representing the median. The minimum/maximum whisker values were calculated as Q1/Q3 ± 1.5 * interquartile range (IQR). Every data point outside is represented as a potential outlier in the form of a dot. Mitochondrial proteins, particularly those of the inner membrane, are significantly depleted. Statistical difference was calculated with a Kolmogorov–Smirnov test comparing the indicated subpopulations with all other proteins. The p values are shown as a measure of statistical significance. The source data for panels (E) and (F) are provided in Dataset EV1. Source data are available online for this figure.

mitochondria with Tom20 (Fig. 5B). Whereas the proteomes of the samples isolated with Rtn1 and Sec63 correlated well, Tom20 and the ER proteins showed distinct protein patterns, as expected (Figs. 5C,D and EV5A–C).

Using the affinity purification procedure allowed us to test, whether the ER-mitochondria contact sites are critical for the accurate protein distribution between both compartments. In Mdm34-depleted cells, many proteins were shifted from the mitochondrial fractions to the affinity-purified ER membranes (Fig. 5D). For a general analysis, we calculated the ER-mitochondria ratio for each protein (i.e., the relative log2 fold change in abundance in the ER (Rtn1/Sec63) vs. mitochondrial fraction) in samples derived from wild type as well as from mutant cells. As shown in Fig. 5E, the ratios generally correlated very well (being close to the diagonal of the graph). However, a distinct set of proteins clearly deviated: The proteins indicated in Fig. 5E were almost exclusively found in the mitochondrial fraction in wild-type samples but had lost their mitochondrial distribution in Mdm34-depleted cells. Thus, the ERMES complex is crucial for the accurate targeting of these proteins to mitochondria (Appendix Fig. S1). Similarly, many mitochondrial proteins were enriched in the ER fraction of Δtom70 samples and, as expected, compartmental identity was most severely compromised in the Mdm34↓/Δtom70 double mutants (Fig. EV5D–F).

The characteristic distribution of Mdm34- and Tom70-dependent proteins identified 84 mitochondrial proteins, which were relatively enriched in the ER and/or depleted from mitochondria in the mutant samples (Dataset EV4; Appendix Fig. S1). 45 of these proteins were inner membrane proteins with hydrophobic transmembrane segments, such as Oxa1, Sdh3, Cox5a and Cox11, and most of the other proteins also contained hydrophobic stretches as part of their sequence. Noteworthy, we did not find representatives of the carrier/SLC25 family (Dataset EV4) in this group, and thus, hydrophobicity alone is apparently not sufficient for an ER association of mitochondrial proteins (Fig. 5F,G). However, we cannot exclude technical reasons, such as low detection efficiencies for the group of carrier proteins.

In summary, upon the depletion of the ERMES subunit Mdm34, a large and rather defined group of mitochondrial proteins was found to be stranded on the ER. This mislocalization on the ER was particularly pronounced for hydrophobic inner membrane proteins. When Mdm34 was depleted in the Δtom70 background, the mitochondrial proteome lost its specific identity, and the affinity-purified fractions showed very similar proteomes, despite the specific isolation scheme of the affinity purification procedure.

## Hydrophobic transmembrane segments determine the contact site dependence of mitochondrial proteins

The observed accumulation of hydrophobic inner membrane proteins on the ER in cells lacking the ER-mitochondria contact sites inspired us to directly test the relevance of Mdm34 and Tom70 for the import of mitochondrial precursor proteins. To this end, we generated semi-intact cells that lacked or contained Mdm34 (by CRISPRi-mediated depletion) and Tom70 (by deletion) and incubated these with radiolabeled precursor proteins of the mitochondrial inner membrane proteins Oxa1 and Cox5a, as well as of the soluble and hydrophilic matrix protein Hsp60 (Fig. 6A,B). Import of the membrane proteins was considerably affected in the double mutants, whereas Hsp60 was efficiently targeted to mitochondria. We next tested, whether the transmembrane domain was relevant for the observed difference. When we used a Cox5a variant in which the transmembrane segment (residues 98 to 113) was replaced by the second hydrophobic transmembrane domain from Oxa1 (Meier et al, 2005), this Cox5a(Oxa1) protein remained contact site-dependent. In contrast, when the transmembrane domain of Cox5a was deleted, the resulting Cox5aΔTM version was efficiently imported into the mitochondria of semi-intact cells even if both Tom70 and Mdm34 were absent. This confirmed that the presence of a single hydrophobic stretch in a membrane protein can be sufficient to render it dependent on contact sites.

The expression of the artificial ER-mitochondria tether did not improve the import of Oxa1 in semi-intact cells from the Δtom70 Mdm34-depleted strain (Appendix Fig. S2A,B). Moreover, import defects for ER-SURF substrates were not only observed in Tom70/Mdm34-depleted strains, but also in combination mutants of Tom70/Mdm12, Djp1/Mdm34, and Lam6/Mdm34 (Appendix Fig. S2C–F), thus in all combinations in which both ER-mitochondria contact sites were compromised. We regard it as unlikely that the import defect is caused by a reduced membrane potential in these strains as even Δcox18 semi-intact cells, which lack cytochrome c oxidase, imported Oxa1 rather efficiently in this assay (Appendix Fig. S2G,H).

Next, we tested whether Mdm34 physically interacts with mitochondrial precursor proteins. We incubated radiolabeled precursors of Oxa1 and Cox5a(Oxa1) with semi-intact cells of wild type or of a mutant in which Mdm34 carried a C-terminal HA tag. After incubation for 5 min, the cells were lysed, and Mdm34-HA was purified on beads carrying antibodies against the HA tag (Fig. 6C; Appendix Fig. S2I). The precursor forms of Oxa1 and Cox5a(Oxa1) were efficiently co-purified with Mdm34-HA,

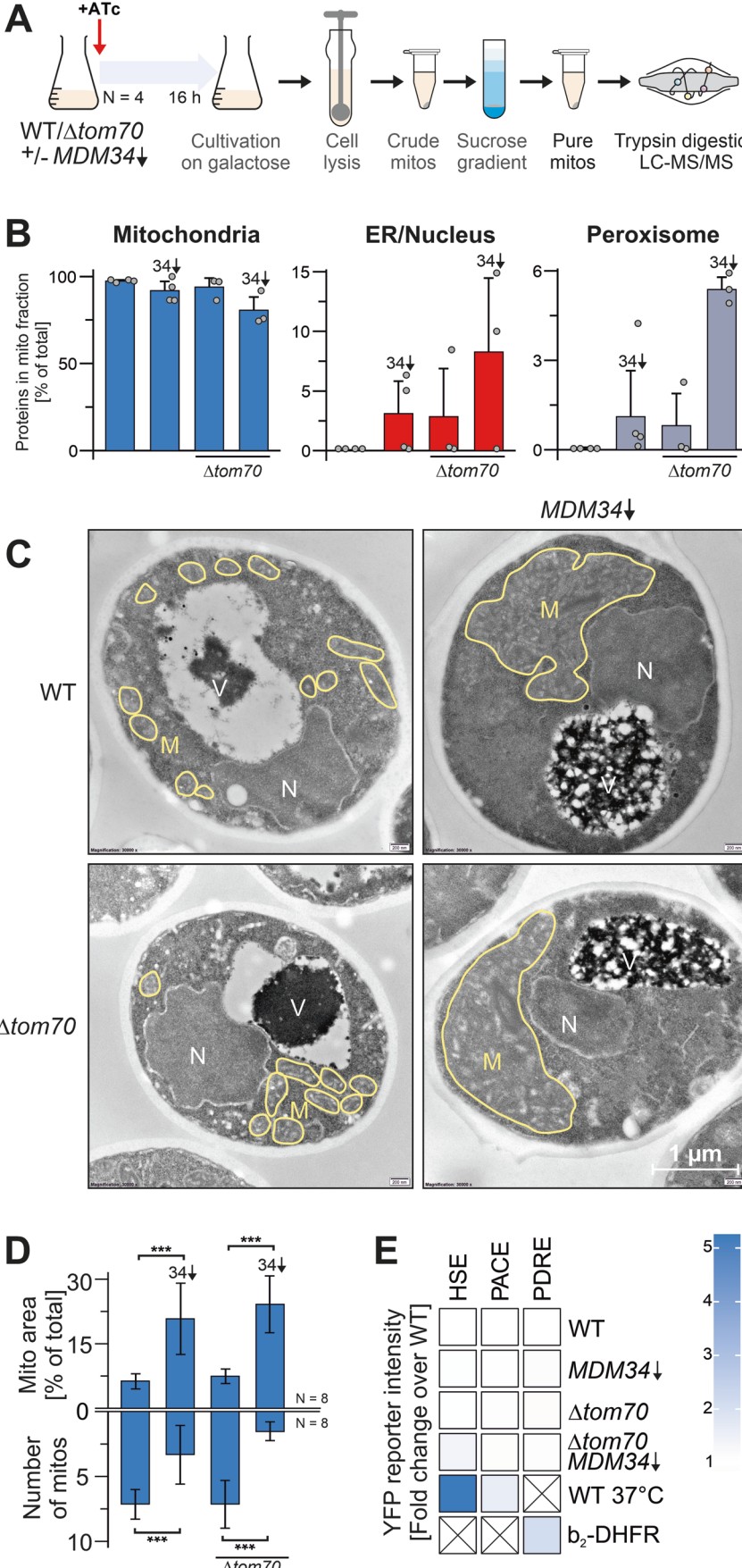

Figure 4. Depletion of ER-mitochondria contact sites changes mitochondrial properties.

(A) Schematic representation of the workflow for the analysis of the proteomic analysis of mitochondrial fractions. Cells were grown in galactose-containing media, and the depletion of *MDM34* was induced for 16 h before mitochondria were isolated. Subsequently, mitochondria were lysed and subjected to LC-MS/MS. (B) Proteins in these fractions were identified and quantified by mass spectrometry as described in the materials and methods. The relative intensities measured of proteins from mitochondria (Morgenstern et al, 2017), the ER, and the nucleus as well as of peroxisomes (Wiederhold et al, 2010) in the "mitochondria" fraction was quantified. Shown are mean values and standard deviations of three (Δ*tom70*) and four (WT) biological replicates. The source data are provided in Dataset EV2. (C, D) Yeast cells were grown to log phase in galactose medium before 960 ng/ml ATc was added for 16 h. Cells were embedded, cut into thin slices, and visualized by transmission electron microscopy. Mitochondrial membranes are indicated in yellow and the number and relative area of mitochondria in the section was quantified from eight different cells per strain. Shown are the mean values and standard deviations of eight samples. Statistical difference was calculated with a student's *t*-test. ***p* value <0.005 M, mitochondria, N, nucleus, V, vacuole. (E) Cells of the indicated strains were grown on galactose in the presence of ATc for 16 h. Cells contained reporters for the gene expression under the control of the heat shock element (HSE), the proteasome-associated control element (PACE), or the pleiotropic drug response element (PDRE). Source data are available online for this figure.

suggesting that ER-SURF substrates directly bind to the ERMES complex. As ERMES is a heterooligomeric complex, the binding might occur via Mdm34 or via another ERMES subunit. Noteworthy, only the precursor but not the mature form of Oxa1 was isolated with ERMES, confirming that preprotein binding occurs prior to their import into the mitochondria.

To assess the relevance of ER-mitochondria contact sites in vivo, we overexpressed Oxa1 in cells from a galactose-inducible promoter for 4 h. We observed that depletion of Mdm34 and deletion of Tom70 resulted in increased amounts of Oxa1 precursor indicative of the impaired import of this membrane protein into mitochondria (Fig. 6D,E). In the double mutant, hardly any mature Oxa1 was observed and a strong synthetic defect in Oxa1 biogenesis was observed. This strongly supports the crucial relevance of ER-mitochondria contact sites for the intracellular targeting of mitochondrial inner membrane proteins. Defects in the mitochondrial lipid composition of the contact site mutants might contribute to the observed changes in the import behavior of their mitochondria.

## Discussion

We developed a CRISPRi-based strategy to rapidly deplete the ERMES subunit Mdm34 and observed that the loss of ERMES resulted in a considerable remodeling of the mitochondrial proteome. Loss of Mdm34 reduced the levels of many mitochondrial proteins, in particular those with hydrophobic segments, and many of these proteins accumulated instead in the ER fractions. This observation is consistent with the idea that ERMES contacts promote the ER-to-mitochondria transfer of precursors on the ER-SURF targeting route (Hansen et al, 2018; Xiao et al, 2021) (Fig. 6F). Interestingly, a recent systematic study with libraries of GFP-tagged mitochondrial proteins reported the accumulation of a number of mitochondrial precursor proteins on the ER upon mitochondrial dysfunction, most of which being hydrophobic membrane proteins (Shakya et al, 2021). Moreover, mitochondrial dysfunction leads to the accumulation of mitochondrial proteins on the ER and induces the unfolded protein response of the ER (Coyne et al, 2023; Knöringer et al, 2023; Sarkar et al, 2022; Xiao et al, 2021). Upon depletion of Mdm34 in Δ*tom70* cells, the affinity-purified ER and mitochondrial fractions partially lost their characteristic protein composition, indicating that the ER-mitochondria contact sites are essential for organellar protein identity. Thus, at least one contact site to the ER is apparently essential for mitochondrial protein biogenesis.

We regard it as unlikely that the accumulation of mitochondrial proteins on the ER is indirectly caused by the altered lipid composition of the Mdm34-depleted cells for several reasons: (1) in the in vitro import experiments with isolated mitochondria, contact sites were not relevant and mitochondria purified from single or double mutants maintained their import capacity; (2) in the semi-intact cell assay, the depletion of Mdm34 also did not impair the import of ER-SURF proteins despite the absence of a lipid-transferring ERMES contact in these cells; (3) however, as soon as Mdm34 and Tom70 were absent, Oxa1 and other membrane proteins were not efficiently imported into mitochondria of semi-intact cells; and (4) a recent study showed that the loss of ERMES reduced the ER-to-mitochondria transfer of lipids; importantly, the overall lipid composition of mitochondria remained almost unaffected unless the mitochondria-vacuole contact site was also lost (John Peter et al, 2022). However, changes in the mitochondrial lipid composition in the contact site mutants might also contribute to the reduced abundance of mitochondrial proteins.

Together, these results suggest that ERMES, in addition to its role in lipid transport, serves a function in the protein transfer from the ER to mitochondria, that is redundant with that of the Tom70-Djp1/Lam6 contact site (Fig. 6F). Whether ERMES thereby interacts with precursor proteins directly or just provides the necessary proximity of the two compartments will have to be analyzed in the future. However, the observation that precursors of ER-SURF substrates were efficiently co-isolated with Mdm34-HA in our semi-intact cell assay indeed suggests a direct role in the ER-to-mitochondria transfer of precursor proteins.

Even though a direct ERMES homolog has not been identified in human cells, a recent study reported a similar ER-to-mitochondria protein transfer by mitochondria-associated ER membranes (MAM), which was facilitated by Tom20 (Lalier et al, 2021). There is good evidence that these ER-mitochondria contact sites are of crucial relevance for organellar identity and function. For example, a recent study showed that in Alzheimer's patients, the accumulation of the Tau protein disturbs ER-mitochondria contacts in neurons which alters the mitochondrial protein composition (Szabo et al, 2023). Interestingly, drugs that stabilize ER-mitochondria contacts showed the potential to restore neuronal homeostasis in an Alzheimer's disease model (Dentoni et al, 2022; Leal et al, 2016).

The ER-SURF-dependence of mitochondrial precursor proteins was most apparent for membrane proteins (Dataset EV4; Fig. 6F). However, we also found a number of soluble proteins that were ER-SURF-dependent; interestingly, most of these proteins contain hydrophobic stretches in their primary structure, which however

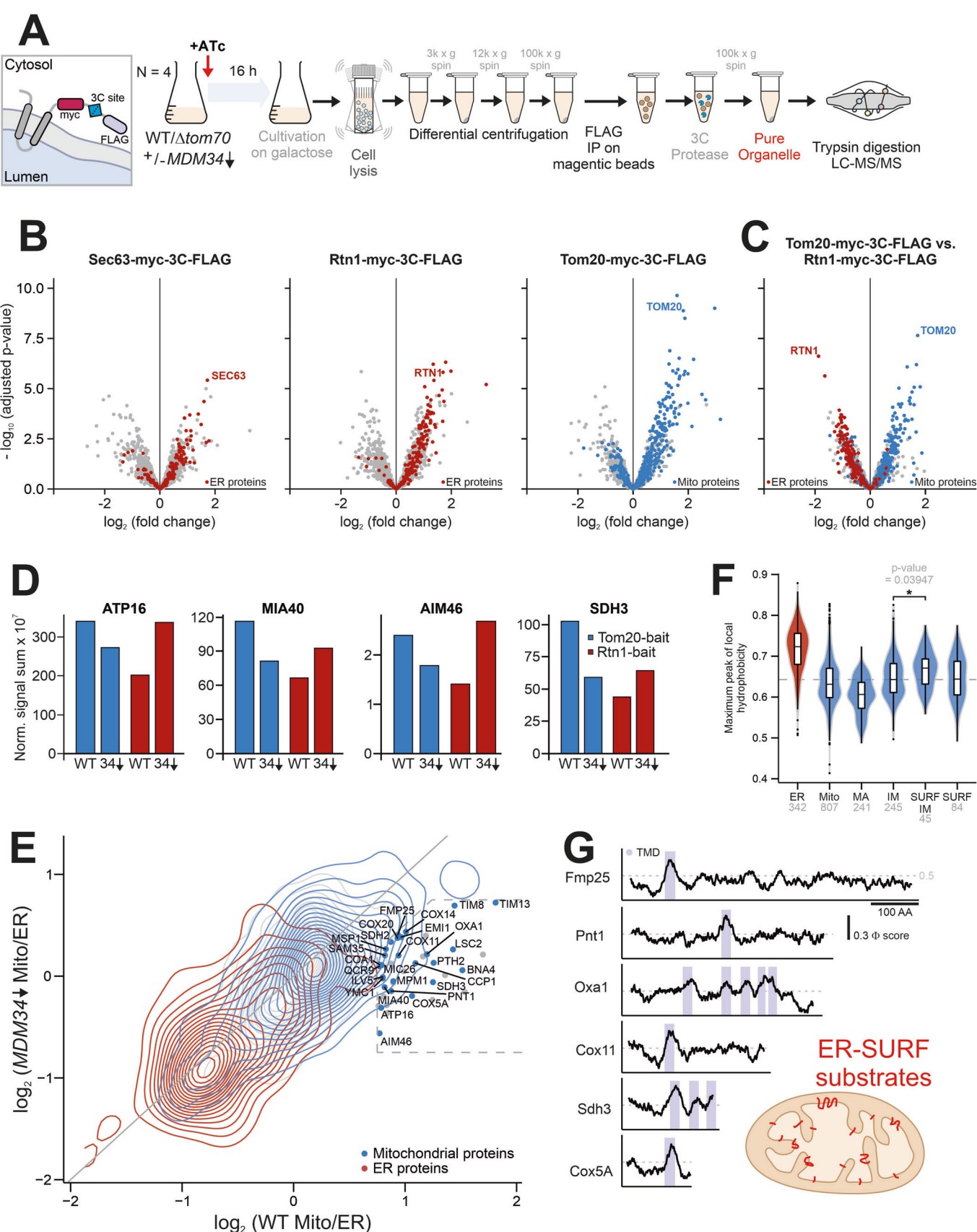

Figure 5. **Many inner membrane proteins accumulate on the ER in the absence of contact sites.**

(A) Schematic representation of the affinity purification of ER and mitochondrial membranes. First, all indicated strains were grown in galactose-containing media, and the depletion of ERMES was induced for 16 h. Afterward, cells were subjected to a crude subcellular fractionation followed by an immunoprecipitation against the flag epitope. The obtained eluates were lysed and subjected to LC-MS/MS. (B, C) Volcano plots of proteomes of the affinity-purified fractions. All measurements are based on three biological replicates and were processed as described in Materials and Methods. For the calculation of fold changes and $p$ values, the limma package within the R programming language was used (Ritchie et al, 2015). (D) Normalized signal intensities for specified proteins in either wild type or Mdm34-depleted cells for Tom20 or Rtn1 pulldowns. The source data for panels (B–D) are provided in Dataset EV3; proteins of mitochondria (Morgenstern et al, 2017) and the ER (Wiederhold et al, 2010) were labeled according to previous publications. (E) Correlation plot showing the log2 fold changes of the ER vs mitochondria fractions from the Rtn1 and Tom20 affinity purification samples from wild type (x-axis) and Mdm34-depleted (y-axis) cells. Samples on the diagonal were unaffected by Mdm34 depletion, whereas samples in the area indicated by the dashed line were found to be enriched on the ER only upon knock-down of Mdm34. The names of these ERMES-dependent mitochondrial proteins are shown. (F) The hydrophobicity scores, based on the Kyte and Doolittle scales (Kyte and Doolittle, 1982), were calculated for 20-residue windows of mitochondrial and ER proteins. The highest value (i.e., the maximum peak of local hydrophobicity) was calculated and is shown for different compartments. Numbers below the different subclasses represent the number of proteins ($n$) within a given subclass. Boxes represent the data range from the first (Q1) to the third quartile (Q3), with the line in the middle representing the median. The minimum/maximum whisker values were calculated as Q1/Q3 ±1.5 * interquartile range (IQR). Every data point outside is represented as a potential outlier in the form of a dot. Please note that ER proteins tend to be more hydrophobic than mitochondrial proteins. Among the mitochondrial proteins, inner membrane proteins, particularly those dependent on the ER-SURF pathway, tend to show an increased hydrophobicity. ER-SURF-dependent proteins were defined as proteins which were shifted from the mitochondrial to the ER fraction (deviation from diagonal >0.5 and log2 (WT EV Mito/ER) >0) upon depletion of Mdm34 or deletion of Tom70. Statistical difference was calculated with a Kolmogorov–Smirnov test comparing the indicated subpopulations. The $p$ values are shown as a measure of statistical significance. (G) Hydrophobicity profiles of several Mdm34-dependent inner membrane proteins. Hydrophobicity ($\Phi$) scores were calculated from 20-residue windows (Kyte and Doolittle, 1982). Transmembrane domains (TMD) are highlighted. AA amino acid residues. Source data are available online for this figure.

are not surface-exposed in their folded structures. We regard it as likely that both the membrane and the soluble proteins are targeted to the ER surface by the signal recognition particle and by chaperones of the ER surface such as Ydj1, Djp1, and Get3, which interact with mitochondrial precursor proteins (Caplan et al, 1992; Costa et al, 2018; Drwesh et al, 2022; Gamerdinger et al, 2015; Hansen et al, 2018; Xiao et al, 2021). Intriguingly, proximity-based ribosome profiling revealed that many mitochondrial proteins are synthesized by ER-bound ribosomes (Jan et al, 2014; Knöringer et al, 2023). It is not known whether this ER association of mitochondrial precursors is an unavoidable mislocalization owing to the fact that the cytosolic targeting machinery erroneously interprets some mitochondrial proteins as ER proteins or, which seems more likely, that eukaryotic cells use the ER as a buffer to absorb potentially harmful precursor proteins to reduce their toxic potential (Coyne et al, 2023; Knöringer et al, 2023; Nowicka et al, 2021; Sutandy et al, 2023).

The contact zone of the ER and mitochondria was recently identified as a hotspot for several cellular quality control processes, such as the degradation of membrane-bound proteins by Ubx2-dependent ER- and mitochondria-associated degradation (Mårtensson et al, 2019; Metzger et al, 2020; Schulte et al, 2023), the formation of autophagosome membranes (Böckler and Westermann, 2014) or the formation of membrane-bound protein aggregates (Grassi et al, 2018; Volgyi et al, 2015; Zhou et al, 2014). It will be exciting to dissect the molecular steps in more detail by which the ER surface promotes the biogenesis of mitochondrial precursor proteins in the future.

## Methods

### Strains and growth conditions

The yeast strains and plasmids used in this study are described in detail in Tables EV1, EV2, respectively. Unless specified, all strains were derived from YPH499 (MATa ura3 lys2 ade2 trp1 his3 leu2).

The strains were grown at 30 °C either in yeast complete medium (YP) containing 1% (w/v) yeast extract, 2% (w/v) peptone, and 2% (w/v) of the respective carbon source or in minimal synthetic medium containing 0.67% (w/v) yeast nitrogen base and 2% (w/v) of the respective carbon source. To induce expression of the gRNA for CRISPRi 960 ng/ml or 240 ng/ml (for microscopic experiments), ATc was added unless indicated otherwise. For induction of GAL1, 0.5% galactose was added to the medium, or cells were pelleted and resuspended in a galactose-containing medium.

### Growth assays and viability tests

For spot analysis, the respective yeast strains were grown in liquid media. Yeast cells equivalent to 0.5 $OD_{600}$ were harvested at the exponential phase. The cells were washed in sterile water and 3 µl of tenfold serial dilutions were spotted on the respective media, followed by incubation at 30 °C. Pictures were taken after different days of the incubation.

Growth curves were performed in a 96-well plate, using the automated ELx808™ Absorbance Microplate Reader (BioTek®). The growth curves started at 0.1 $OD_{600}$ and the $OD_{600}$ was measured every 10 min for 72 h at 30 °C. The mean of technical triplicates was calculated and plotted in R.

### YFP reporter assays

The PACE-YFP, HSE-YFP, and PDRE-YFP reporter genes were integrated into the LEU2 locus of the yeast genome. Cells were induced by the addition of 960 ng/ml ATc for 16 h in galactose-containing media. As positive controls the empty vector sample shifted to 37 °C for 16 h (PACE and HSE) and 4 h induction of $b_2$-DHFR (PDRE) were used. 4 $OD_{600}$ of cells were harvested by centrifugation (12,000 × $g$, 5 min, RT) and resuspended in 400 µl $H_2O$. About 100 µl of the cell suspension were transferred to flat-bottomed black 96-well imaging plates (BD Falcon, Heidelberg, Germany) in technical triplicates. Cells were sedimented by gentle

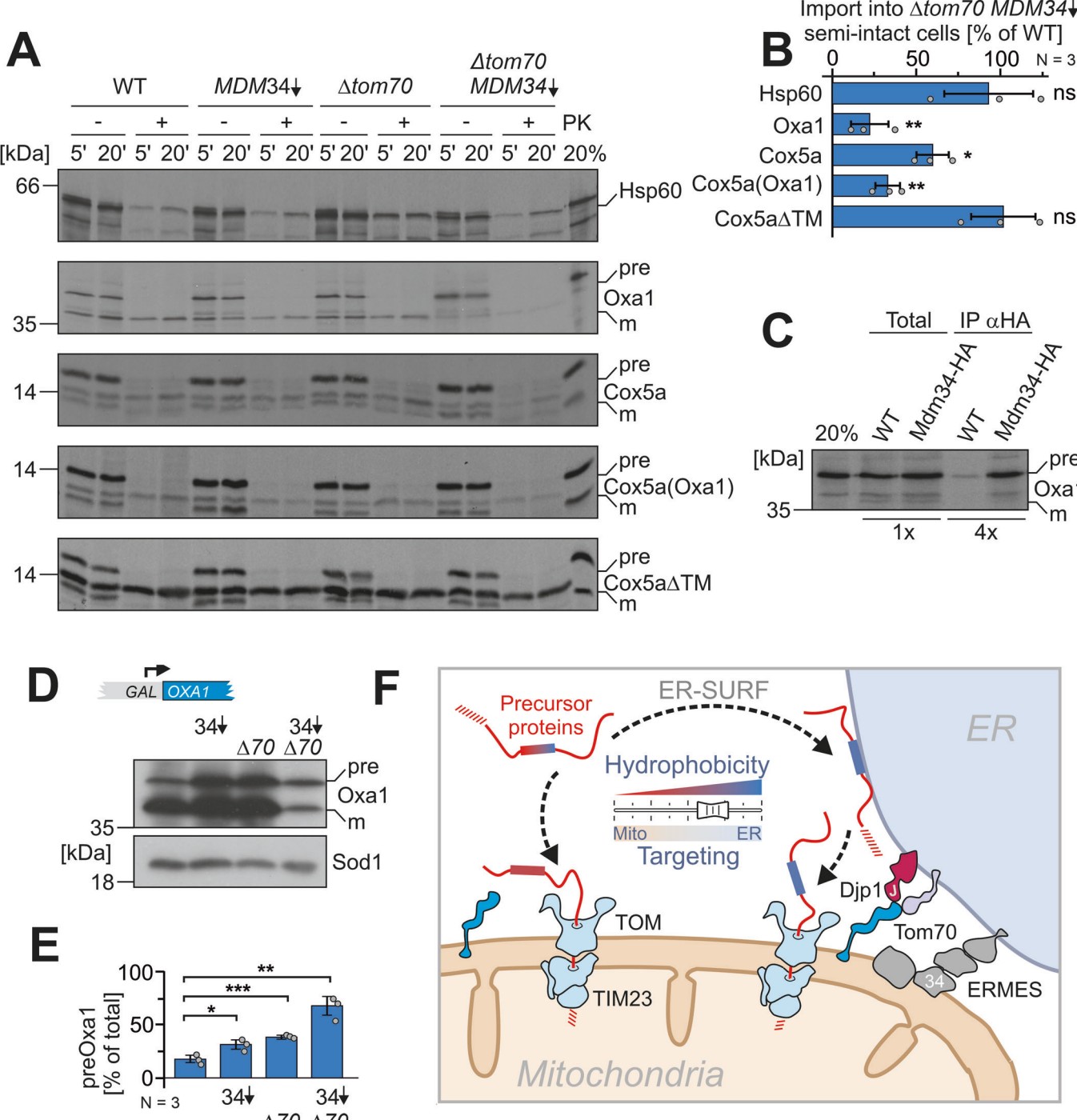

spinning (30 g, 5 min, RT), and fluorescence (excitation 497 nm, emission 540 nm) was measured using a ClarioStar Fluorescence plate reader (BMG-Labtech, Offenburg, Germany). The corresponding wild-type strain not expressing YFP was used for background subtraction of autofluorescence. Fluorescence intensities were normalized to the value obtained from the wild-type empty vector control in each of three independent biological replicates.

## Cell lysates

For whole-cell lysates, yeast strains were cultivated in liquid media to mid-log phase. 2 $OD_{600}$ were harvested by centrifugation ($12,000 \times g$, 5 min) and resuspended in 100 µl reducing loading buffer. Cells were transferred to screw-cap tubes containing 1 mm glass beads. Cell lysis was performed using a FastPrep-24 5G homogenizer (MP Biomedicals, Heidelberg, Germany) with three

◀  **Figure 6.  The hydrophobicity of internal segments determines the ER-SURF-dependence.**

(**A**) Radiolabeled precursor proteins were synthesized in reticulocyte lysate and incubated with semi-intact cells of the indicated strains. After 5 and 20 min, aliquots were taken before samples were treated with proteinase K (PK). Hsp60 is a hydrophilic matrix protein. Oxa1 contains five transmembrane segments. Cox5a is a single-spanning inner membrane protein. For Cox5a(Oxa1), the transmembrane domain of Cox5a was replaced by the second transmembrane domain of Oxa1. For Cox5aΔTM, the transmembrane domain was deleted so that the protein is now translocated into the matrix (Meier et al, 2005). (**B**) The experiment shown in A was repeated three times from independent semi-intact cell preparations (biological replicates) and the signals of the imported proteins in wild type and Mdm34-depleted Δtom70 cells after 20 min of incubation were quantified. Shown are mean values and standard deviations. Statistical difference was calculated with a student's *t*-test. ns non significant, *$p$ value <0.05, **$p$ value <0.01. (**C**) Radiolabeled Oxa1 precursor was incubated with semi-intact cells of wild type and Mdm34-HA expressing cells. After 5 min, cells were lysed. The extract was incubated with HA-specific antibodies coupled to protein A Sepharose beads. Beads were washed before bound proteins were eluted with sample buffer and subjected to SDS-PAGE. The total samples correspond to 25% of what is shown in the IP (immune precipitation) samples. (**D**, **E**) Oxa1 was expressed from a strong *GAL* promoter in the lactate-containing medium by the addition of 0.5% galactose for 4 h in the indicated mutants. Western blots with HA-specific antibodies showed signals for the precursor (pre) and mature (m) Oxa1 protein. Signals from three independent (biological) replicates were quantified. Mean values and standard deviations are shown. Statistical difference was calculated with a student's *t*-test. Statistical significance was assigned as follows: *$p$ value <0.05, **$p$ value <0.01, ***$p$ value <0.005. (**F**) Schematic representation of the ER-supported protein import into mitochondria. Proteins with hydrophobic patches show an increased tendency to associate with the ER surface, but can efficiently be passed on to the mitochondrial import machinery. For the productive protein transfer from the ER to mitochondria, the presence of the ER-mitochondria contact sites (ERMES and Tom70-Djp1/Lam6) are crucial. See discussion for details. Source data are available online for this figure.

cycles of 30 s, speed 8.0 m/s, 120 s breaks, and glass beads). Lysates were boiled at 96 °C for 3 min and stored at −20 °C until further use. Equal amounts were resolved via SDS-PAGE.

## Antibodies

The antibodies against Sod1 and Djp1 were raised in rabbits using recombinant purified proteins. The antibodies against Rip1 and Mdj1 were kindly gifted by Thomas Becker (University of Freiburg, Germany). The horseradish-peroxidase coupled HA antibody was ordered from Roche (Anti-HA-Peroxidase, High Affinity 3F10, #12 013 819 001). The secondary antibodies were ordered from Biorad (Goat Anti-Rabbit IgG (H + L)-HRP Conjugate #172-1019). Antibodies were diluted in 5% (w/v) nonfat dry milk-TBS (Roth T145.2) with the following dilutions: anti-Sod1 1:500, anti-Djp1 1:2000 anti-Rip1 1:750, anti-Mdj1 1:125, anti-Rabbit 1:10,000.

## Analysis of mRNA levels by qRT-PCR

For total RNA extraction, yeast strains were cultivated in synthetic media to the mid-log phase. 4 $OD_{600}$ of cells were harvested and RNA was extracted using the RNeasy Mini Kit (Qiagen) in conjunction with the RNase-Free DNase Set (Qiagen) according to the manufacturer's instructions. Yield and purity of the obtained RNA was determined with a Spectrophotometer/Fluorometer DS-11 FX+ (DeNovix). About 500 ng RNA were reverse transcribed into cDNA using the qScript cDNA Synthesis Kit (Quanta Biosciences) according to the manufacturer's instructions. To measure relative mRNA levels, the iTaq Universal SYBR Green Supermix (BioRad) was used with 2 µl of a 1:10 dilution of the cDNA sample. For assessment of *HAC1*$_{spliced}$ mRNA, the Luna Universal Probe One-Step RT-qPCR Kit (NEB) was used with 2 µl of a 50 ng/µl RNA sample. Measurements were performed in technical triplicates with the CFX96 Touch Real-Time PCR Detection System (BioRad). Calculations of the relative mRNA expressions were conducted following the $2^{-\Delta\Delta Ct}$ method (Livak and Schmittgen, 2001). For normalization, the housekeeping genes *TFC1* or *ACT1* were used due to their stability (Teste et al, 2009). See Table EV3 for primer sequences.

## Isolation of mitochondria

For the isolation of mitochondria, cells were grown in rich or selective galactose media to the mid-log phase. Cultures for CRISPRi samples

were additionally treated with ATc (960 ng/ml final) for 16 h. Cells were harvested (2000×*g*, 5 min, RT) in the exponential phase. After a washing step, cells were treated for 10 min with 2 ml per g wet weight MP1 buffer (100 mM DTT, 10 mM Tris pH unadjusted) at 30 °C. After washing with 1.2 M sorbitol, yeast cells were resuspended in 6.7 ml per g wet weight MP2 buffer (20 mM KPi buffer pH 7.4, 1.2 M sorbitol, 3 mg per g wet weight zymolyase 20 T from Seikagaku Biobusiness) and incubated for 1 h at 30 °C. Spheroplasts were collected via centrifugation at 4 °C and resuspended in ice-cold homogenization buffer (13.4 ml/g wet weight) (10 mM Tris pH 7.4, 1 mM EDTA pH 8, 0.2% fatty acids free bovine serum albumin (BSA), 1 mM PMSF, 0.6 M sorbitol). Spheroplasts were disrupted by 10 strokes with a cooled glass potter. Cell debris was removed via centrifugation at 1500×*g* for 5 min. The supernatant was centrifuged for 12 min at 12,000×*g* for 10 min to collect mitochondria. Mitochondria were resuspended in 1 ml of ice-cold SH buffer (0.6 M sorbitol, 20 mM Hepes pH 7.4). The mitochondria were diluted to a protein concentration of 10 mg/ml.

## Isolation of semi-intact cells

Cells were grown to the mid-log phase. 250 $OD_{600}$ of cells were harvested by centrifugation (700×*g*, 7 min, room temperature). The cell pellet was resuspended in 25 ml SP1 buffer (10 mM DTT, 100 mM Tris pH unadjusted) and incubated for 10 min at 30 °C in a shaker. Cells were pelleted (1000×*g*, 5 min, room temperature), resuspended in 6 ml SP2 buffer (0.6 M sorbitol, 1x YP, 0.2% glucose, 50 mM KPi pH 7.4, 3 mg/g wet weight zymolyase) and incubated at 30 °C for 30–60 min. Spheroblasts were collected, resuspended in 40 ml of SP3 buffer (1x YP, 1% glucose, 0.7 M sorbitol), and incubated for 20 min at 30 °C shaking. After centrifugation (1000 g, 5 min, 4 °C), spheroblasts were washed two times with 20 ml of ice-cold permeabilization buffer (20 mM Hepes pH 6.8, 150 mM KOAc, 2 mM Mg-Acetate, 0.4 M sorbitol). The pellet was resuspended in 1 ml permeabilization buffer containing 0.5 mM EGTA and 100 µl aliquots were slowly frozen over liquid nitrogen for 30 min. A detailed procedure with pictures of how cells should be slowly frozen was published before (Laborenz et al, 2019).

## Protein import into mitochondria

The TNT® Quick Coupled Transcription/Translation Kit from Promega was for the synthesis of $^{35}$S-methionine labeled proteins in reticulocyte lysate. About 50 µg mitochondria were taken in import buffer (500 mM

sorbitol, 50 mM Hepes pH 7.4, 80 mM KCl, 10 mM Mg(OAc)$_2$, and 2 mM KH$_2$PO$_4$), 2 mM ATP, and 2 mM NADH and incubated for 10 min at 30 °C. The import reaction was started by the addition of 1% (v/v) reticulocyte lysate. Samples were taken after the indicated time points, and the reaction was stopped by a 1:10 dilution in ice-cold SH buffer supplemented with 100 µg/ml proteinase K. The samples were incubated on ice for 30 min to remove precursors which were not imported. The protease treatment was stopped by the addition of 2 mM PMSF. The samples were centrifuged for 15 min at 25,000×$g$ and 4 °C. The mitochondria were washed with 500 µl SH/KCl-buffer (0.6 M sorbitol, 20 mM HEPES/KOH pH 7.4, 150 mM KCl) and 2 mM PMSF. The mitochondria were reisolated by centrifugation for 15 min at 25,000×$g$ and 4 °C, resuspended in sample buffer, and resolved via SDS-PAGE.

## Import into semi-intact cells

Semi-intact cells were thawed on ice and the OD$_{600}$ was measured in 1.2 M sorbitol. Semi-intact cells of an OD$_{600}$ 0.2 were used per reaction. Semi-intact cells were added to a mixture of B88 buffer, 2 mM ATP, 2 mM NADH, 5 mM creatine phosphate, and 100 µg/ml creatine phosphatase. The radiolabeled lysate was added, and the mixture was first incubated for 10 min on ice to allow the cells to take up the lysate before the suspensions were shifted to 30 °C for 5 and 20 min to allow the import of proteins into mitochondria. The import reactions were stopped by a 1:10 dilution in ice-cold B88 buffer containing 2 mM CCCP, with or without 100 µg/ml proteinase K. After an incubation of 30 min on ice, 2 mM PMSF were added. Semi-intact cells were reisolated by centrifugation (4000×$g$, 5 min, 4 °C), washed with B88, 2 mM PMSF, and pelleted by centrifugation (10 min, 16,000×$g$, 4 °C). Finally, the pellet was resuspended in the sample buffer and resolved via SDS-PAGE.

## Import into semi-intact cells followed by immunoprecipitation

The import into semi-intact cells of wild type and Mdm34-HA was carried out as described above for 5 min. The import reactions were stopped by a 1:10 dilution in ice-cold B88 buffer containing 2 mM CCCP. Afterward, semi-intact cells were reisolated by centrifugation (4000×$g$, 5 min, 4 °C) and lysed by the addition of 100 µl lysis buffer (10 mM Tris pH 7.5, 150 mM NaCl, 0.5 mM EDTA, 0.5% Triton X-100). About 20 µl of this lysate were used as total samples. The rest was incubated with 30 µl equilibrated Protein A beads, 3 µl HA antibody (Sigma #H3663), and filled up to a total volume of 200 µl with lysis buffer. The reaction was incubated for 1 h at 4 °C tumbling end-over-end. Subsequently, the beads were settled via centrifugation (20 s at 2000×$g$) and washed thrice with 800 µl wash buffer I (50 mM Tris pH 7.5, 150 mM NaCl, 5% glycerol, 0.05% Triton X-100) and twice with 500 µl wash buffer II (50 mM Tris pH 7.5, 150 mM NaCl, 5% glycerol). Proteins were eluted from the beads by the addition of 30 µl of reducing sample buffer. All samples were boiled for 3 min at 96 °C and resolved via SDS-PAGE.

## Sample preparation and mass spectrometric identification of proteins

For the quantitative comparison of proteomes of Δ*tom70* and WT cells, carrying the CRISPRi EV or CRISPRi *MDM34*, were

induced for 8 and 24 h with ATc. 10 OD$_{600}$ of cells were harvested at each time point by centrifugation (12,000×$g$, 5 min) and snap-frozen in liquid nitrogen and stored at −80 °C. Cells lysates were prepared in lysis buffer (50 mM Tris pH 7.5, 2% (w/v) SDS, Tablets mini EDTA-free protease inhibitor (Roche)) using a FastPrep-24 5 G homogenizer (MP Biomedicals, Heidelberg, Germany) with three cycles of 30 s, speed 8.0 m/s, 120 s breaks, glass beads). Lysates were boiled for 5 min at 96 °C and centrifuged (16,000×$g$, 2 min, 4 °C). Protein concentrations were determined using the Pierce BCA Protein Assay (Thermo Scientific, #23225). About 20 µg of each lysate were subjected to an in-solution tryptic digest using a modified version of the Single-Pot Solid-Phase-enhanced Sample Preparation (SP3) protocol. Here, lysates were added to Sera-Mag Beads (Thermo Scientific, #4515-2105-050250, 6515-2105-050250) in 10 µl 15% formic acid and 30 µl of ethanol. The binding of proteins was achieved by shaking for 15 min at room temperature. SDS was removed by four subsequent washes with 200 µl of 70% ethanol. Proteins were digested with 0.4 µg of sequencing grade modified trypsin (Promega, #V5111) in 40 µl 20 mM Hepes pH 8.4 in the presence of 1.25 mM TCEP and 5 mM chloroacetamide (Sigma-Aldrich, #C0267) overnight at room temperature. Beads were separated, washed with 10 µl of an aqueous solution of 2% DMSO, and the combined eluates were dried down. In total, three biological replicates were prepared ($n = 3$). Peptides were reconstituted in 10 µl of H$_2$O and reacted with 80 µg of TMT10plex (Thermo Scientific, #90111) label reagent dissolved in 4 µl of acetonitrile for 1 h at room temperature. Excess TMT reagent was quenched by the addition of 4 µl of an aqueous solution of 5% hydroxylamine (Sigma, 438227). Peptides were mixed to achieve a 1:1 ratio across all TMT-channels. Mixed peptides were desalted on home-made StageTips containing Empore C$_{18}$ disks (Rappsilber et al, 2007) and subjected to an SCX fractionation on StageTips into 3 fractions, followed by additional cleanup on C$_{18}$ StageTips. The resulting fractions were then analyzed by LC-MS/MS on a Q Exactive HF (Thermo Scientific) as previously described.

For mass spectrometry of sucrose-purified mitochondria, first crude mitochondria were prepared as described above from either WT or Δ*tom70* harboring an empty vector or a knockdown vector. In total, four/three biological replicates were prepared for WT ($n = 4$) and Δ*tom70* ($n = 3$), respectively. For purification, a four-step sucrose gradient was poured into SW41 rotor tubes. The gradient consisted of 1.5 ml 60% sucrose, 4 ml 32% sucrose, 1.5 ml 23% sucrose and 1.5 ml 15% sucrose (w/v) in EM buffer (20 mM MOPS-KOH pH 7.2, 1 mM EDTA). About 5 mg of crude mitochondria were loaded onto the gradient and centrifuged at 134,000×$g$ for 1 h at 2 °C. The pure mitochondria were collected from the 32/60% interface and diluted with two volumes of SEM buffer (20 mM MOPS-KOH pH 7.2, 1 mM EDTA, 0.6 M sorbitol). Afterwards, mitochondria were reisolated at 15,000×$g$ for 15 min at 2 °C. This pellet was resuspended in SEM and the protein concentration was determined with the Bradford assay (Bradford, 1976). About 100 µg of pure mitochondria were lysed in 100 µl lysis buffer (50 mM Tris pH 7.5, 2% (w/v) SDS, Tablets mini EDTA-free protease inhibitor (Roche)) by boiling for 10 min at 96 °C. Protein concentrations were determined using the Pierce BCA Protein Assay (Thermo Scientific, #23225). About 20 µg of each lysate were subjected to an in-solution tryptic digest using a modified version of the Single-Pot Solid-Phase-enhanced Sample Preparation (SP3) protocol. Here, lysates were added to Sera-Mag Beads (Thermo

Scientific, #4515-2105-050250, 6515-2105-050250) in 10 µl 15% formic acid and 30 µl of ethanol. The binding of proteins was achieved by shaking for 15 min at room temperature. SDS was removed by four subsequent washes with 200 µl of 70% ethanol. Proteins were digested with 0.4 µg of sequencing grade modified trypsin (Promega, #V5111) in 40 µl 20 mM Hepes pH 8.4 in the presence of 1.25 mM TCEP and 5 mM chloroacetamide (Sigma-Aldrich, #C0267) overnight at room temperature. Beads were separated, washed with 10 µl of an aqueous solution of 2% DMSO, and the combined eluates were dried down. Peptides were reconstituted in 20 µl of $H_2O$, acidified to pH <2 with Tri-flouracetic acid, and desalted with 3x $C_{18}$ StageTips. Samples were dried down in speed-vac and resolubilized in 9 µl buffer A (0.1% formic acid in MS grade water) and 1 µl buffer A$^*$ (2% acetonitrile, 0.1% tri-flouracetic acid in MS grade water). The samples were analyzed by LC-MS/MS on a Q Exactive HF (Thermo Scientific) as previously described (Sridharan et al, 2019).

For IP mass spectrometry of purified organelles, a modified protocol from Reinhard et al, was used (Reinhard et al, 2023). WT and Δtom70 strains carrying a genomic bait-tag on Rtn1, Sec63, Tom20 or no tag, and in turn, each harboring an empty vector or a knockdown vector were grown in galactose medium containing ATc to induce repression of MDM34. For each strain, biological quadruplicates (n = 4) were prepared (64 samples in total). After 16 h, 100 $OD_{600}$ cells were harvested by centrifugation (5000×g for 5 min) and resuspended in 1 ml SEH buffer (20 mM HEPES pH 7.4, 1 mM EDTA, 0.6 M Sorbitol, 1x cOmplete™ Tablets mini EDTA-free protease inhibitor [Roche]). Resuspended yeast cells were transferred into screw-cap tubes containing 500 µl glass beads. Cell lysates were prepared using a FastPrep-24 5 G homogenizer (MP Biomedicals, Heidelberg, Germany) with ten cycles of 15 s, speed 5.0 m/s, and 45 s breaks. Afterward, lysates were subjected to subcellular fractionation via differential centrifugation. Samples were centrifuged for 5 min at 3300×g, for 20 min at 12,000×g, and for 1 h at 100,000×g. After each step, the pellet was discarded and only the last pellet (P100), containing heavy membranes, was collected, resuspended in 100 µl SEH buffer, snap-frozen in liquid $N_2$, and stored at −80 °C until further use. About 900 µl of IP buffer (20 mM HEPES pH 7.4, 1 mM EDTA, 100 mM NaCl) was added to P100 prior to the IP. Diluted samples were mixed with 50 µl of Pierce Anti-DYKDDDDK Magnetic Agarose from Thermo. The beads were equilibrated by washing 200 µl of beads slurry once with 1 ml PBS pH 7.4 and twice with 1 ml of wash buffer (20 mM HEPES pH 7.4, 1 mM EDTA, 75 mM NaCl). Samples were bound to Anti-DYKDDDDK beads for 2 h at 4 °C tumbling end-over-end. IP Samples were briefly centrifuged (1 min at 2500×g) and the supernatant was discarded by using a magnetic rack. Beads were washed 3x with 1 ml wash buffer, and finally, 400 µl elution buffer (PBS pH 7.4, 0.5 mM EDTA, 0.03 mg/ml HRV-3C protease [Sigma-Aldrich #SAE0045]) was added onto the beads. Organelles were eluted for 2 h at 4 °C tumbling end-over-end, the beads were separated with a magnetic rack and eluted organelles were transferred to a fresh microtube. Afterwards, intact organelles were pelted via centrifugation for 2 h at 200,000×g and the resulting pellet was resuspended in 35 µl lysis buffer (6 M GdmCl, 10 mM TCEP, 40 mM CAA, 100 mM Tris pH 8.5). Organelles were lysed by boiling for 10 min at 96 °C. For protein digestion, first lysed samples were diluted 1:10 with digestion buffer (10% ACN, 25 mM Tris pH 8.5), next Trypsin and LysC were added in a 1:50 ratio and

the reaction was incubated overnight at 37 °C. The next day fresh Trypsin was added in a 1:100 ratio for 30 min at 37 °C. The pH of samples was adjusted to pH <2 with tri-flouracetic acid. Desalting/reversed-Phase cleanup with 3x SDB-RPS StageTips. Samples were dried down in speed-vac and resolubilized in 12 µl buffer A$^{++}$ (0.1% formic acid, 0.01% tri-flouracetic acid in MS grade water). The samples were analyzed by LC-MS/MS on a Q Exactive HF (Thermo Scientific) as previously described (Sridharan et al, 2019).

Briefly, peptides were separated using an Easy-nLC 1200 system (Thermo Scientific) coupled to a Q Exactive HF mass spectrometer via a Nanospray-Flex ion source. The analytical column (50 cm, 75 µm inner diameter (NewObjective) packed in-house with C18 resin ReproSilPur 120, 1.9 µm diameter Dr. Maisch) was operated at a constant flow rate of 250 nl/min. A 3 h gradient was used to elute peptides (Solvent A: aqueous 0.1% formic acid; Solvent B: 80% acetonitrile, 0.1% formic acid). Peptides were analyzed in positive ion mode applying with a spray voltage of 2.3 kV and a capillary temperature of 250 °C. For TMT-labeled peptides, MS spectra were acquired in profile mode with a mass range of 375–1.400 m/z using a resolution of 120.000 [maximum fill time of 80 ms or a maximum of 3e6 ions (automatic gain control, AGC)]. Fragmentation was triggered for the top 15 peaks with charge 2–8 on the MS scan (data-dependent acquisition) with a 30 s dynamic exclusion window (normalized collision energy was 32). Precursors were isolated with a 0.7 m/z window and MS/MS spectra were acquired in profile mode with a resolution of 60,000 (maximum fill time of 100 ms, AGC target of 1e5 ions, fixed first mass 100 m/z).

For label-free peptides, MS spectra were acquired in profile mode with a mass range of 300 1650 m/z using a resolution of 60.000 [maximum fill time of 20 ms or a maximum of 3e6 ions (automatic gain control, AGC)]. Fragmentation was triggered for the top 15 peaks with charge 2–8 on the MS scan (data-dependent acquisition) with a 20 s dynamic exclusion window (normalized collision energy was 28). Precursors were isolated with a 1.4 m/z window and MS/MS spectra were acquired in profile mode with a mass range of 200 to 2000 m/z and a resolution of 15,000, maximum fill time of 80 ms, AGC target of 1e5 ions.

## Analysis of mass spectrometry data

Peptide and protein identification and quantification was done using the MaxQuant software (version 1.6.10.43) (Cox and Mann, 2008; Cox et al, 2011; Tyanova et al, 2016) and a Saccharomyces cerevisiae proteome database obtained from Uniprot. For TMT-labeled peptides, 10plex TMT was chosen in Reporter ion MS2 quantification, up to two tryptic miss-cleavages were allowed, protein N-terminal acetylation and Met oxidation were specified as variable modifications and Cys carbamidomethylation as fixed modification. The "Requantify" and "Second Peptides" options were deactivated. The false discovery rate was set at 1% for peptides, proteins, and sites, minimal peptide length was seven amino acids. For label-free data, the LFQ normalization algorithm and second peptides was enabled. Match between run was applied within each group of replicates. The false discovery rate was set at 1% for peptides, proteins, and sites, minimal peptide length was seven amino acids.

The output files of MaxQuant were processed using the R programming language. Only proteins that were quantified with at least two unique peptides were considered for the analysis. Moreover, only proteins that were identified in at least two out of three MS runs

per replicate were kept. A total of 3550 proteins for the whole-cell proteome, a total of 1624 proteins for the pure mitochondria, and a total of 3045 for the organelle-IP passed the quality control filters. Raw signal sums were cleaned for batch effects using limma (Ritchie et al, 2015) and further normalized using variance stabilization normalization (Huber et al, 2002). Proteins were tested for differential expression using the limma package for the indicated comparison of strains. A reference list of yeast mitochondrial proteins was used (Morgenstern et al, 2017). For results of the analysis of all MS experiments, see Datasets EV1–EV3.

## Fluorescence microscopy

For microscopy, cells were grown to mid-log phase in galactose-containing media if not indicated otherwise, and 1 OD was harvested via centrifugation. Cell pellets were resuspended in 30 µl of PBS. About 3–5 µl were pipetted onto a glass slide and covered with a cover slip. Manual microscopy was performed using a Leica Dmi8 Thunder Imager. Images were acquired using an HC PL APO100x/1,44 Oil UV objective with Immersion Oil Type A 518 F. Excitation of 510 and 575 nm were used for mNeongreen and mScarlet-I, respectively. All mitochondrial images were taken as Z-stacks. Image analysis was done with the LAS X software, and further processing of images was performed in Fiji/ImageJ.

For time-lapse imaging, first a glass slide with an agar pad had to be prepared. Therefore, synthetic liquid media supplemented with 2% galactose was mixed with 1.5% (w/v) agarose, boiled in a microwave, and 180 µl was pipetted into a single cavity slide (42410010 Karl Hecht Assistent). The mixture was flattened by pressing a second glass slide on top of the hot mixture, orthogonal to the cavity slide. Just before adding cells onto the agar pad, the upper glass slide was removed. Cells were grown and harvested as above but resuspended in liquid minimal media. About 3–5 µl were pipetted onto the agar pad and covered with a cover slip. Images were taken every 5 min while the microscope incubation chamber was heated to 30 °C.

## Electron microscopy and immuno-electron microscopy

Aliquots of semi-intact cells were thawed on ice and fixed in 2% glutaraldehyde, and 3% formaldehyde overnight at 4 °C. The samples were processed as described (Prescianotto-Baschong and Riezman, 2002). Sections were quenched with 50 mM glycine in PBS for 15 min and washed with PBS 3x for 10 min. Grids were blocked with PBST + 2% BSA for 15 min and labeled with monoclonal mouse anti-porin antibodies (16G9E6BC4, cat # 459500, Thermo Fisher) in a 1:100 dilution in PBST-BSA. A secondary goat anti-mouse IgG 10 nm gold conjugated antibody (BBInternational SKU#. BA GAM40) 1:100 in PBST-BSA for 2 h was used. Sections were washed 5 × 5 min in PBS, fixed with 1% glutaraldehyde in PBS for 5 min, and washed in $H_2O$. Proteins were visualized using 2% uranyl acetate stain for 10 min and lead citrate (Reynold's) for 1 min. Sections were analyzed in a Philips CM100 electron microscope.

## Data availability

The mass spectrometry proteomics data (see also Tables EV1–EV3) have been deposited to the ProteomeXchange Consortium via the PRIDE (Perez-Riverol et al, 2019) partner repository with the dataset identifier shown below. The datasets are available via the following URLs: Proteomics Datasets: URL. Dataset 1: Enrichment of ER/Mitos https://www.ebi.ac.uk/pride/archive/projects/PXD044362. Dataset 2: Isolated Mitos https://www.ebi.ac.uk/pride/archive/projects/PXD044368. Dataset 3: Total proteome https://www.ebi.ac.uk/pride/archive/projects/PXD044379.

## Peer review information

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

## Acknowledgements

We thank Vera Nehr and Sabine Knaus for technical assistance, Benoit Kornmann (University of Oxford, UK) for a plasmid for expression of the Tom70-GFP-Ubc6 tether, and Robert Ernst (Saarland University, Homburg, Germany) for the pJR2 plasmid for the affinity purification of organelles. This project was supported by grants from the European Research Council (ERC 101052639 MitoCyto to JMH), the Deutsche Forschungsgemeinschaft (HE2803/10-1 and GRK2737-STRESSistance to JMH), the Landesforschungsinitiative Rheinland-Pfalz BioComp (to JMH), and the Swiss National Science Foundation (310030_185127 to AS).

## Author contributions

**Christian Koch**: Conceptualization; Data curation; Software; Investigation; Visualization; Writing—original draft. **Svenja Lenhard**: Data curation; Writing—review and editing. **Markus Räschle**: Data curation; Supervision; Writing—review and editing. **Cristina Prescianotto-Baschong**: Data curation; Writing—review and editing. **Anne Spang**: Project administration; Writing—review and editing. **Johannes M Herrmann**: Conceptualization; Supervision; Funding acquisition; Investigation; Writing—original draft; Project administration.

## Funding

## Disclosure and competing interests statement

The authors declare no competing interests.

# Expanded View Figures

**Figure EV1.   The ERMES contact site is critical for protein targeting via ER-SURF.**

(**A**, **B**) The protein levels of Djp1 and of control proteins in wild type, Δ*djp1* and Δ*mdm34* strains were analyzed by Western blotting and quantified. Panel **B** shows the mean values and standard deviations of three biological replicates. Statistical difference was calculated with a student's *t*-test. Statistical significance was assigned as follows: ***$p$ value <0.005. (**C**, **D**) Schematic representation of the Oxa1-Ura3 reporter assay. Normal import of this reporter leads to the depletion of Ura3 from the cytosol and uracil auxotrophy. Impaired import of this reporter restores uracil prototrophy and allows for growth on plates lacking uracil. Cells of the indicated strains were grown to log phase in glucose medium before tenfold serial dilutions were dropped onto plates containing or lacking uracil. (**E–G**) Radiolabeled proteins were synthesized in reticulocyte lysate in the presence of $^{35}$S-methionine and incubated with semi-intact cells derived from the indicated strains. After 5 or 20 min, the cells were isolated, treated without or with proteinase K (PK) for 30 min on ice, and subjected to SDS-PAGE and autoradiography. 20% of the radioactive protein used per import reaction (time point) was loaded for comparison. Precursor and mature forms are indicated as pre and m. Source data are available online for this figure.

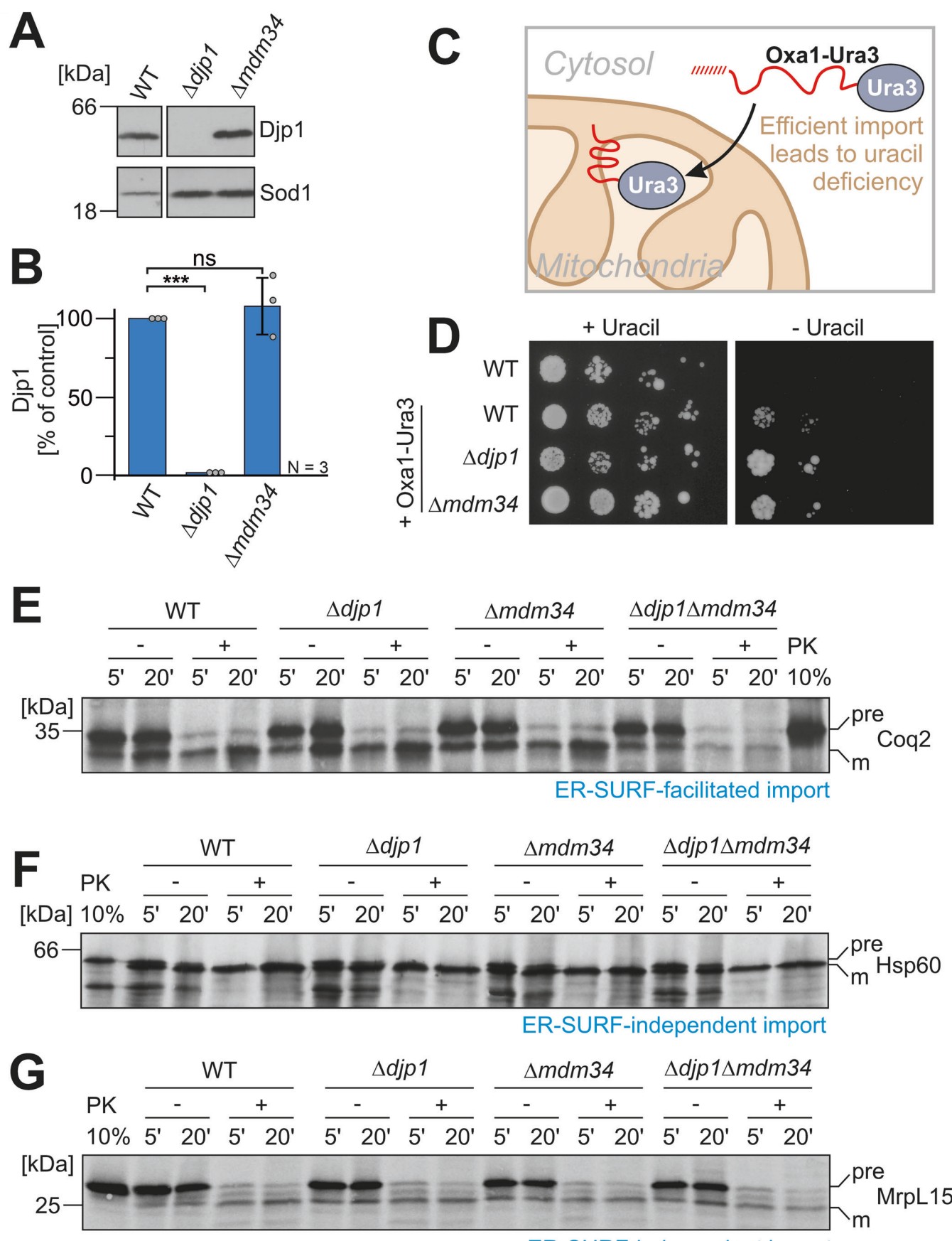

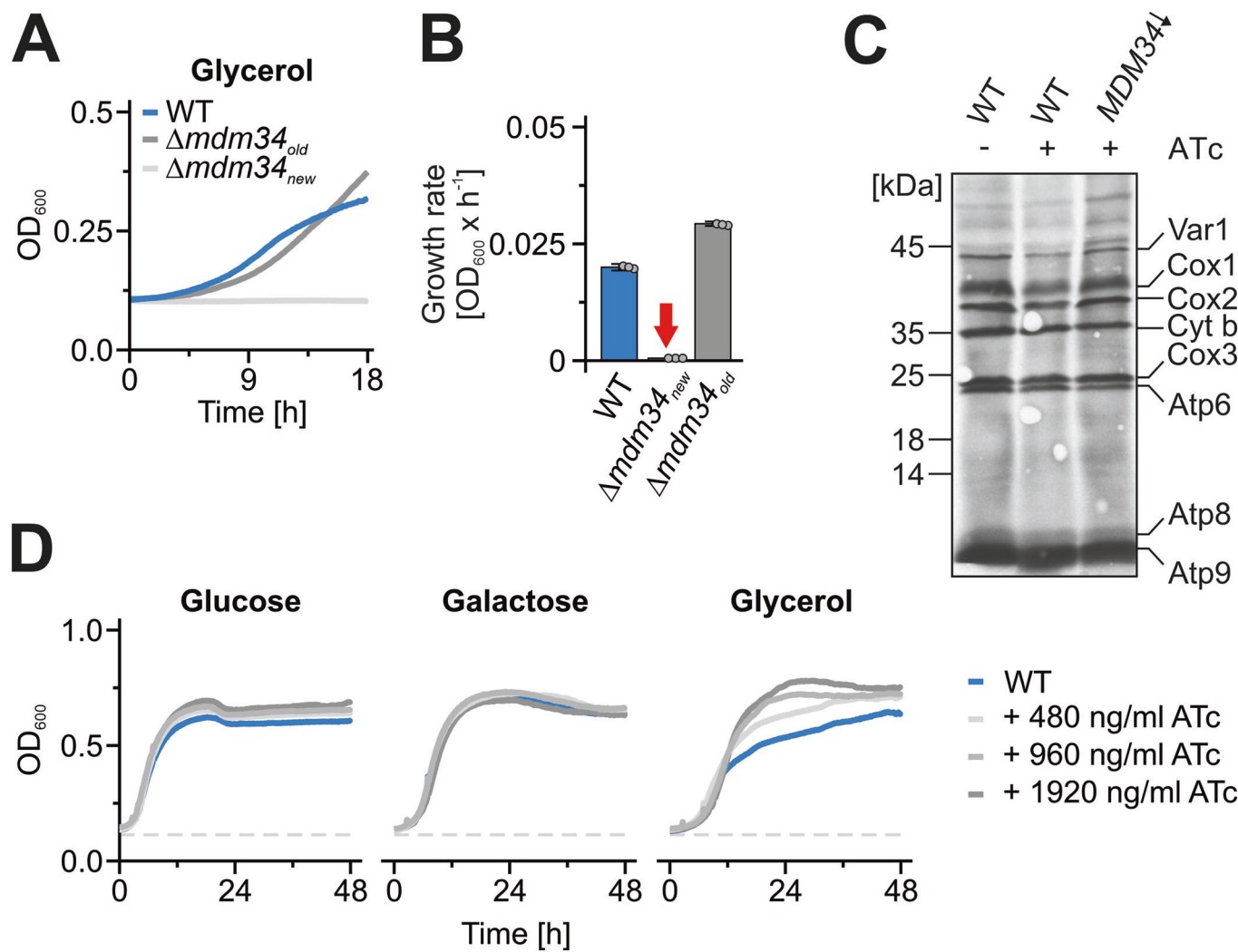

**Figure EV2.  The deletion of ERMES contact sites lead to transient growth defects.**

(A, B) The indicated strains were grown in galactose medium to log phase and used to inoculate cultures in glycerol media. Cells were grown at 30 °C under constant agitation. Cell growth was continuously monitored. Growth rates were determined by calculation of the slope of the curve in the log phase. The graphs show the mean values of three technical replicates. (C) Indicated strains were grown in galactose medium and expression of Mdm34 was suppressed by the addition of ATc for 16 h. Mitochondrial translation products were radiolabeled for 15 min with $^{35}$S-methionine in the presence of cycloheximide to inhibit cytosolic translation. Radiolabeled proteins were visualized by SDS-PAGE and autoradiography. (D) The indicated strains were grown in galactose medium to log phase and used to inoculate cultures in either glucose, galactose, or glycerol media containing varying concentrations of ATc. Cells were grown at 30 °C under constant agitation. Cell growth was continuously monitored. Source data are available online for this figure.

                                   

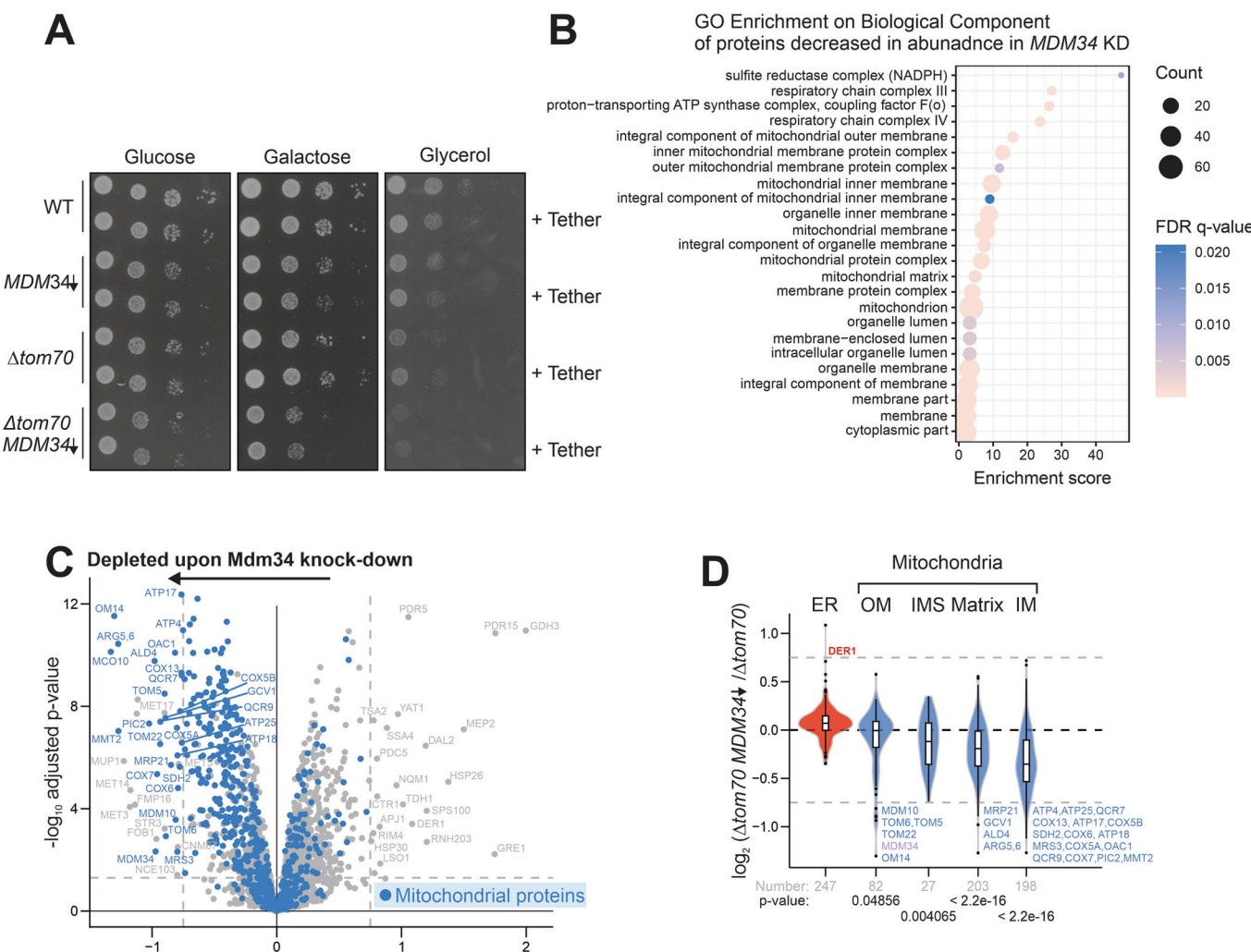

**Figure EV3. Loss of ERMES and Tom70 leaves a strong footprint on the mitochondrial proteome.**

(**A**) Cells of the indicated strains were grown to log phase in galactose medium and depletion of Mdm34 was induced by the addition of 960 ng/ml of ATc before tenfold serial dilutions were dropped onto plates with the indicated carbon sources. (**B**) The whole-cell proteomes of wild type and Mdm34-depleted cells were measured and further analyzed by gene ontology (GO) enrichment. Proteins with smaller than −0.5 log$_2$ fold change were used as target set and analyzed by using the GOrilla tool (http://cbl-gorilla.cs.technion.ac.il/) with all quantified proteins as background. The top results with a false discovery rate [FDR] <5% are shown. (**C**) Comparison of the proteomes of Δ*tom70* and Mdm34-depleted cells 24 h after ATc addition. Mitochondrial proteins (Morgenstern et al, 2017) were indicated in blue. (**D**) The violin plot shows the ratio of protein abundance (log2-fold enrichment scores) in Δ*tom70* relative to Mdm34-depleted cells. Numbers below the different subclasses represent the number of proteins (*n*) within a given subclass. Boxes represent the data range from the first (Q1) to the third quartile (Q3), with the line in the middle representing the median. The minimum/maximum whisker values were calculated as Q1/Q3 ± 1.5 * interquartile range (IQR). Every data point outside is represented as a potential outlier in the form of a dot. Mitochondrial proteins, particularly those of the inner membrane, are significantly depleted. Statistical difference was calculated with a Kolmogorov–Smirnov test comparing the indicated subpopulations with all other proteins. The *p* values are shown as a measure of statistical significance. Source data are available online for this figure.

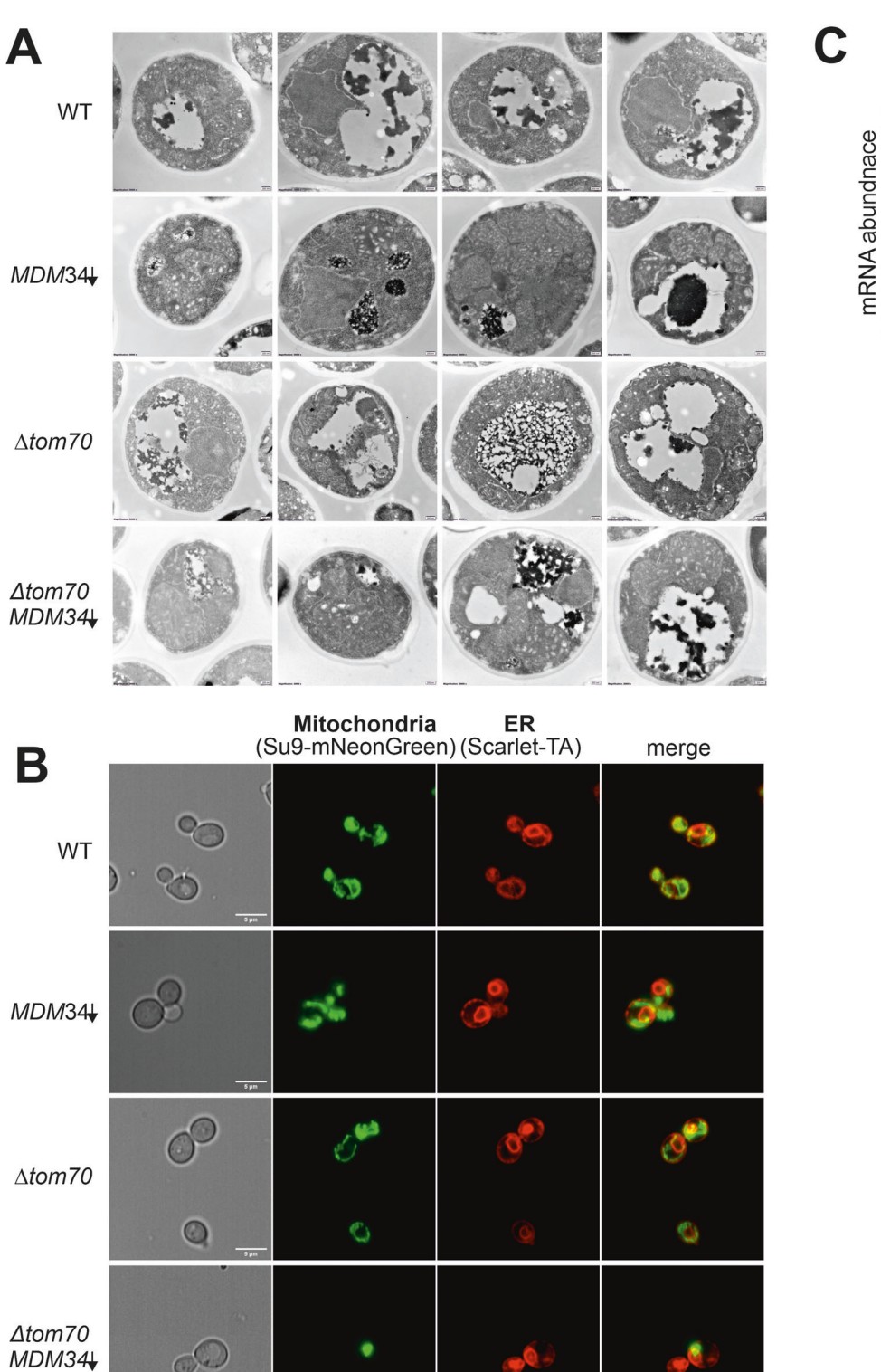

◄ **Figure EV4. Loss of ERMES strongly affects mitochondrial morphology.**

(**A**) Yeast cells were grown to log phase in galactose medium before 960 ng/ml ATc was added for 16 h. Cells were embedded, cut into thin slices, and visualized by transmission electron microscopy. (**B**) The indicated strains expressing a matrix-targeted mNeonGreen (Su9-mNeonGreen) and an ER-targeted mScarlet (Scarlet-TA) were grown to log phase in galactose medium before 960 ng/ml ATc was added for 16 h. Afterwards, cells were harvested and imaged using a Leica Dmi8 Thunder imager. (**C**) Cells of the indicated strains were grown on galactose in the presence of ATc for 16 h. The induction of the unfolded protein response (UPR) was measured by qPCR based on *HAC1* splicing. Wild-type strains treated for 1 h with either DTT or tunicamycin served as positive controls. Source data are available online for this figure.

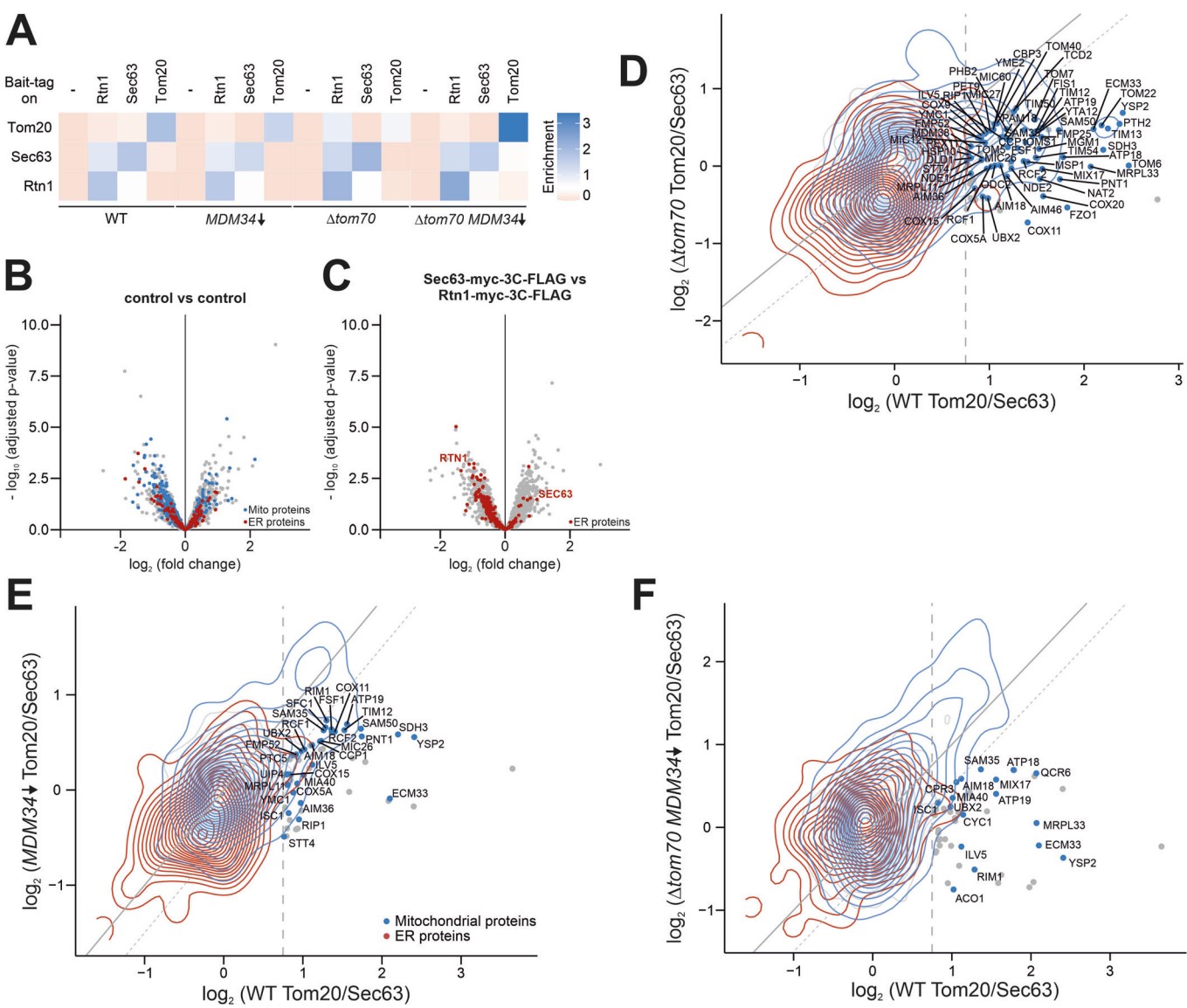

**Figure EV5. Proteomic analysis of ERMES and Tom70-deficient cells identifies ER-SURF clients.**

(A) Specific enrichment scores (difference in log2 fold change) for Tom20 and the ER proteins Sec63 and Rtn1 in the affinity-purified fractions were calculated to validate the selective recovery of the bait proteins. Data show mean values of three biological replicates. (B, C) Volcano plots of the proteomic data of the affinity-purified fractions. All measurements are based on three biological replicates and were processed as described in Materials and Methods. For the calculation of fold changes and *p* values, the limma package within the R programming language was used (Ritchie et al, 2015). (D–F) Correlation plot showing the log2 fold changes of the ER vs mitochondria fractions from the Sec63 and Tom20 affinity purification samples from wild type (x-axis) and Δ*tom70* (D), Mdm34-depleted (E), and Δ*tom70* Mdm34-depleted (F) (y-axis) cells. Samples on the diagonal were unaffected by Mdm34 depletion, whereas samples in the area indicated by the dashed line were classified as putative ER-SURF clients. The names of these mitochondrial proteins are shown.

