## [Peer Review File · EMBO Reports]

The ER-SURF pathway uses ER-mitochondria contact sites for protein targeting to mitochondria

Christian Koch, Svenja Lenhard, Markus Räschle, Cristina Prescianotto-Baschong, Anne Spang, and Johannes Herrmann

Corresponding author(s): Johannes Herrmann (hannes.herrmann@biologie.uni-kl.de)

Review Timeline:

Submission Date:	31st Aug 23
Editorial Decision:	16th Oct 23
Revision Received:	18th Dec 23
Editorial Decision:	8th Feb 24
Revision Received:	14th Feb 24
Accepted:	20th Feb 24

Transaction Report:

Dear Dr. Herrmann

Thank you for the submission of your research manuscript to our journal. We have now received the full set of referee reports that is copied below.

Given the supportive and constructive comments, we would like to invite you to revise your manuscript with the understanding that the referee concerns must be fully addressed and their suggestions taken on board. Please address all referee concerns in a complete point-by-point response. Acceptance of the manuscript will depend on a positive outcome of a second round of review. It is EMBO Reports policy to allow a single round of revision only and acceptance or rejection of the manuscript will therefore depend on the completeness of your responses included in the next, final version of the manuscript.

We realize that it is difficult to revise to a specific deadline. In the interest of protecting the conceptual advance provided by the work, we recommend a revision within 3 months (January 16, 2023). Please discuss the revision progress ahead of this time with the editor if you require more time to complete the revisions.

I am also happy to discuss the revision further via e-mail or a video call, if you wish.

*****IMPORTANT NOTE:

We perform an initial quality control of all revised manuscripts before re-review. Your manuscript will FAIL this control and the handling will be delayed IN CASE the following APPLIES:

- 1) A data availability section providing access to data deposited in public databases is missing. If you have not deposited any data, please add a sentence to the data availability section that explains that.
- 2) Your manuscript contains statistics and error bars based on $n=2$. Please use scatter blots in these cases. No statistics should be calculated if $n=2$.

When submitting your revised manuscript, please carefully review the instructions that follow below. Failure to include requested items will delay the evaluation of your revision. *****

- 1) a .docx formatted version of the manuscript text (including legends for main figures, EV figures and tables). Please make sure that the changes are highlighted to be clearly visible.
- 2) individual production quality figure files as .eps, .tif, .jpg (one file per figure). Please download our Figure Preparation Guidelines (figure preparation pdf) from our Author Guidelines pages <https://www.embopress.org/page/journal/14693178/authorguide> for more info on how to prepare your figures.
- 3) a .docx formatted letter INCLUDING the reviewers' reports and your detailed point-by-point responses to their comments. As part of the EMBO Press transparent editorial process, the point-by-point response is part of the Review Process File (RPF), which will be published alongside your paper.
- 4) a complete author checklist, which you can download from our author guidelines (<<https://www.embopress.org/page/journal/14693178/authorguide>>). Please insert information in the checklist that is also reflected in the manuscript. The completed author checklist will also be part of the RPF.
- 5) Please note that all corresponding authors are required to supply an ORCID ID for their name upon submission of a revised manuscript (<<https://orcid.org/>>). Please find instructions on how to link your ORCID ID to your account in our manuscript tracking system in our Author guidelines (<<https://www.embopress.org/page/journal/14693178/authorguide#authorshipguidelines>>)
- 6) We replaced Supplementary Information with Expanded View (EV) Figures and Tables that are collapsible/expandable online. A maximum of 5 EV Figures can be typeset. EV Figures should be cited as "Figure EV1, Figure EV2" etc... in the text and their

respective legends should be included in the main text after the legends of regular figures.

7) The Data Availability section (placed after Materials & Method) should follow the model below (see also <<https://www.embopress.org/page/journal/14693178/authorguide#dataavailability>>). Please note that the Data Availability Section is restricted to new primary data that are part of this study.

Data availability

Additional information on source data and instruction on how to label the files are available <<https://www.embopress.org/page/journal/14693178/authorguide#sourcedata>>.

10) Figure legends and data quantification:

- the name of the statistical test used to generate error bars and P values,
 - the number (n) of independent experiments (please specify technical or biological replicates) underlying each data point,
 - the nature of the bars and error bars (s.d., s.e.m.)
- If the data are obtained from n {less than or equal to} 5, show the individual data points in addition to the SD or SEM.
- If the data are obtained from n {less than or equal to} 2, use scatter blots showing the individual data points.

11) Our journal encourages inclusion of *data citations in the reference list* to directly cite datasets that were re-used and obtained from public databases. Data citations in the article text are distinct from normal bibliographical citations and should directly link to the database records from which the data can be accessed. In the main text, data citations are formatted as follows: "Data ref: Smith et al, 2001" or "Data ref: NCBI Sequence Read Archive PRJNA342805, 2017". In the Reference list, data citations must be labeled with "[DATASET]". A data reference must provide the database name, accession number/identifiers and a resolvable link to the landing page from which the data can be accessed at the end of the reference. Further instructions are available at <<https://www.embopress.org/page/journal/14693178/authorguide#referencesformat>>.

12) All Materials and Methods need to be described in the main text. We would encourage you to use 'Structured Methods', our new Materials and Methods format. According to this format, the Materials and Methods section should include a Reagents and Tools Table (listing key reagents, experimental models, software and relevant equipment and including their sources and

relevant identifiers) followed by a Methods and Protocols section in which we encourage the authors to describe their methods using a step-by-step protocol format with bullet points, to facilitate the adoption of the methodologies across labs.

More information on how to adhere to this format as well as downloadable templates (.doc or .xls) for the Reagents and Tools Table can be found in our author guidelines: <

<https://www.embopress.org/page/journal/14693178/authorguide#manuscriptpreparation>>.

<<https://www.embopress.org/doi/10.15252/msb.20178071>>.

13) As part of the EMBO publication's Transparent Editorial Process, EMBO Reports publishes online a Review Process File to accompany accepted manuscripts. This File will be published in conjunction with your paper and will include the referee reports, your point-by-point response and all pertinent correspondence relating to the manuscript.

Yours sincerely,

Referee #1:

The study by Koch et al. addresses the molecular mechanisms of how mitochondrial precursor proteins can be transferred from the surface of the endoplasmic reticulum to their mitochondrial destination. In an elegant series of experiments, the authors demonstrate that ER-mitochondria contact sites play an important role in this process. They elucidate that the multifunctional mitochondrial import receptor Tom70 and the ER-mitochondria encounter structure ERMES function in two parallel pathways for the transfer of mitochondrial precursor proteins from the ER to mitochondria, preferentially precursors of inner membrane proteins and some matrix proteins with hydrophobic stretches.

This is a remarkable study of exquisite quality. The leading laboratories of Herrmann and Spang join their efforts in solving several crucial technical problems for the analysis of mutant cells and protein sorting in semi-intact cells and provide important novel insight in early steps of mitochondrial protein biogenesis. The paper is written very well and illustrated by excellent figures with very helpful cartoons outlining the main messages of the paper.

I just have a few minor points:

1. Page 4: 'Single deletions of either Tom70 or ERMES are rather well tolerated; however, double deletions are inviable on all carbon sources.'

Provide a reference for this statement also in the Introduction.

2. Page 10 and further places: Check if Reinhard et al., 2023, or Reinhard et al., 2022, should be cited (see also List of References on page 32).

3. Page 14: 'primary sequence' should be replaced by 'primary structure'.

Referee #2:

Mitochondrial protein trafficking is essential for the maintenance of normal mitochondrial functions. Herrmann's group previously

reported the ER-SURF pathway, which functions as a productive, backup protein transport route to deliver mitochondrial precursor proteins to mitochondria, but via the ER surface. However, except for the involvement of Dj1 in the ER, little was known about the molecular mechanism of the operation of this pathway. Here, Koch et al. report that ER-mitochondria contacts, including ERMES and Tom70-Lam6 sites, are actively involved in the ER-SURF process. The findings and interpretation are provocative as well as inspiring and could have a significant impact on the current understanding of the protein trafficking.

(1) The most crucial point about the authors' claim is whether the observed effects arising from the loss of the contact sites (deletion of Mdm34 or Tom70) are indeed direct consequences. The authors stated in the discussion that this was not due to perturbation in the cellular lipid composition, but loss of inter-organelle contacts. However, the presented experimental evidence may not be sufficiently convincing. Although the authors stated, "the overall lipid composition of mitochondria remained almost unaffected unless the mitochondria-vacuole contact site was also lost (John Peter et al, 2022)", another study (Tamura et al. JBC 287, 15205-15218 (2012)) reported that cellular lipid profiles are affected by the loss of ERMES (decreased cardiolipin level and increased phosphatidylserine level). The authors should test, for example, if the loss of Mdm31, which would cause similar alteration in lipid profiles without affecting inter-organelle contacts, affects the ER-SURF pathway or not. In addition, the dominant positive mutation of Vps13 (D716H), which suppresses the effect of the loss of ERMES on lipid profiles (Lang et al., JCB 210, 883-890 (2015)), can be used, as well.

(2) The authors depleted only Mdm34 as a test to analyze the effect of the loss of ERMES on protein transport and said, "tethering of the two organelles as the artificial Tom70-GFP-Ubc6 tether construct, which bridges mitochondria and ER membranes, did not mitigate the import defect observed in the semi-intact cell assay". However, I am wondering why the authors did not test the loss of other ERMES subunits to rule out the possibility that the observed effect is specific to Mdm34, not the ERMES. Therefore, they should at least test the depletion of another ERMES subunit such as Mdm12. Mdm12 can be presumably depleted quickly by auxin-inducible degron.

Specific points.

Figure 1B - Cellular localization of the accumulated Oxa1 precursor in the absence of Mdm34 should be shown.

Fig. 1D - The EM images did not convincingly indicate that intracellular organization or more specifically, organelle contacts, were not affected in semi-intact cells. The ER, mitochondria, and ER-mitochondria contacts should be analyzed by fluorescent microscopy.

Fig. 1H, I - The effects of Tether on the semi-intact cell import in the absence of Dj1 and Mdm34 are not clear, so they should be quantified. The gel may show that the import defects were mildly recovered by Tether?

Figure 2 - The time course of the decrease in the ER-mitochondria contact sites had better be shown in the Mdm34 knockdown experiment.

Fig. 2 - On the basis of Fig. 2B, the authors said that they used Glu media for further analyses, but the plates for Fig. 2E appear to contain Gal (see page 26, line 2 from the bottom).

Figs. 3, 4, and 5 - Fig. 3 showed the data taken for Glu media, but Figs. 4 and 5 showed those taken for Gal media? The authors should explain the reason for using different media in these experiments.

Fig. 3F - The authors said that, in Mdm34 depleted cells, the amounts of matrix and IM proteins were affected while those of OM and IMS proteins were not affected. However, this could simply reflect the decreased membrane potential across the IM due to altered lipid profiles (pointed out above).

Fig. 4A and B - The possible changes in the lipid profiles in each organelle (pointed out above) could affect the organelle densities, thereby leading to contamination of mitochondrial proteins in other organelle fractions.

Fig. 4C - Organelle shapes are not well seen in these EM pictures. The authors should check the organelle shapes by fluorescence microscopy.

Methods - The used media should be described in detail (only the carbon sources were described). Are they synthetic media or not? How much is the ATc concentration?

Referee #3:

The manuscript by Herrmann et al. describes that the mitochondria-ER contact sites mediated by ERMES and Tom70 are critically important for the localization of the mitochondrial proteins utilizing the ER-SURF pathway. The pathway has been recently identified as a way to mitochondria for a subset of mitochondrial inner membranes proteins, such as Oxa1, that involves the surface of the ER as a platform on the way to mitochondria.

This study identifies ERMES and Tom70 as two parallel routes, by which the ER-SURF substrates are transferred to mitochondria. Both, ERMES and Tom70 are involved in creation of independent contact sites between the ER and mitochondria. Dysfunction of both results in a strong impairment of protein import, changes in membrane signatures and the accumulation of mitochondria precursor proteins on the surface of the ER.

Major points

- It is important to show direct interactions between mitochondrial precursors using ER-SURF with the components of ERMES or other contact sites specific components. Presence or absence of direct interactions will lead to different interpretations on the

requirements of the contact sites and the proteins, which form them. Do they provide the space regulation or constitute a direct step of precursors passing from the ER to the mitochondrial translocases.

- Some of the described effects should be repeated in the absence of Lam6 and Djp to exclude a contribution from the multiple involvements of Tom70 (beyond the ER-mito contact sites)

Minor comments

- The proteomic data should be analyzed and discussed from the perspective of a hypothetical specificity of two different contacts sites. Lack of specificity argues for the indirect architectural role of contacts sites proteins fulfilling the space requirement for the ER-SURF pathway

- Fig 4C, S4A: the labeling in the figure legend is missing.

Point-by-point reply to reviewer comments

Referee #1:

The study by Koch et al. addresses the molecular mechanisms of how mitochondrial precursor proteins can be transferred from the surface of the endoplasmic reticulum to their mitochondrial destination. In an elegant series of experiments, the authors demonstrate that ER-mitochondria contact sites play an important role in this process. They elucidate that the multifunctional mitochondrial import receptor Tom70 and the ER-mitochondria encounter structure ERMES function in two parallel pathways for the transfer of mitochondrial precursor proteins from the ER to mitochondria, preferentially precursors of inner membrane proteins and some matrix proteins with hydrophobic stretches.

This is a remarkable study of exquisite quality. The leading laboratories of Herrmann and Spang join their efforts in solving several crucial technical problems for the analysis of mutant cells and protein sorting in semi-intact cells and provide important novel insight in early steps of mitochondrial protein biogenesis. The paper is written very well and illustrated by excellent figures with very helpful cartoons outlining the main messages of the paper.

I just have a few minor points:

We thank the referee for these wonderful comments. We addressed her/his specific requests as described in the following:

1. Page 4: 'Single deletions of either Tom70 or ERMES are rather well tolerated; however, double deletions are inviable on all carbon sources.' Provide a reference for this statement also in the Introduction.

We added references to the statement about the synthetic lethality of combined TOM70/ERMES deletions. The synthetic lethal interaction has been reported before by Jodi Nunnari's and by our labs (Murley et al. 2015 JCB 209, 539-548; Backes et al. 2021 Cell Reports 35, 108936).

2. Page 10 and further places: Check if Reinhard et al., 2023, or Reinhard et al., 2022, should be cited (see also List of References on page 32).

We now consistently cite the recently published study Reinhard et al. 2023. Thank you for the comment.

3. Page 14: 'primary sequence' should be replaced by 'primary structure'.

We corrected this.

Referee #2:

Mitochondrial protein trafficking is essential for the maintenance of normal mitochondrial functions. Herrmann's group previously reported the ER-SURF pathway, which functions as a productive, backup protein transport route to deliver mitochondrial precursor proteins to mitochondria, but via the ER surface. However, except for the involvement of Djpl in the ER, little was known about the molecular mechanism of the operation of this pathway. Here, Koch et al. report that ER-mitochondria contacts, including ERMES and Tom70-Lam6 sites, are actively involved in the ER-SURF process. The findings and interpretation are provocative as well as inspiring and could have a significant impact on the current understanding of the protein trafficking.

We are grateful for this supportive evaluation and addressed the specific points of the referee as follows:

(1) The most crucial point about the authors' claim is whether the observed effects arising from the loss of the contact sites (deletion of Mdm34 or Tom70) are indeed direct consequences. The authors stated in the discussion that this was not due to perturbation in the cellular lipid composition, but loss of inter-organelle contacts. However, the presented experimental evidence may not be sufficiently convincing. Although the authors stated, "the overall lipid composition of mitochondria remained almost unaffected unless the mitochondria-vacuole contact site was also lost (John Peter et al, 2022)", another study (Tamura et al. JBC 287, 15205-15218 (2012)) reported that cellular lipid profiles are affected by the loss of ERMES (decreased cardiolipin level and increased phosphatidylserine level). The authors should test, for example, if the loss of Mdm31, which would cause similar alteration in lipid profiles without affecting inter-organelle contacts, affects the ER-SURF pathway or not. In addition, the dominant positive mutation of Vps13 (D716H), which suppresses the effect of the loss of ERMES on lipid profiles (Lang et al., JCB 210, 883-890 (2015)), can be used, as well.

We agree with the referee that changes in the mitochondrial lipid composition, as consequence of the loss of contact sites, will influence the mitochondrial proteome.

It is therefore important that our study shows that the ER-mitochondria contacts are of direct relevance for the protein transfer. This is, why we had used the newly established import assay with semi-intact cells in which we can monitor the targeting of precursor proteins to and import into mitochondria. Since we only observe the import defect in in vitro experiments with semi-intact cells (Fig. 1E) but not with isolated mitochondria (Fig. 1F), we regard it as unlikely that the lipid composition explains the import defect. Moreover, the proteomics experiments show that most matrix, IMS and OM proteins are present in normal levels, but proteins with hydrophobic transmembrane domains are retained on the ER. However, we agree with the referee that the loss of contact sites will alter the mitochondrial lipid composition. This has been well documented in excellent studies by the laboratories of Benoit Kornmann, Toshiya Endo, Jodi Nunnari, Thomas Langer, Adam Hughes and others.

We followed the suggestion of the referee and tested mutants lacking Mdm31. However, these mutants had severely compromised mitochondria with strong import defects. In vitro, mitochondria or semi-intact cells isolated from this strain were not import-competent at all. Still, this mutant tolerated the loss of Tom70 well and no synthetic growth defect was observed. Thus, this phenotype is very different to that of ERMES mutants which are synthetic lethal with $\Delta tom70$. We show these results here for inspection by the referee.

We asked Benoit Kornmann for the VPS13(D716H) suppression plasmid but unfortunately were not able to receive this plasmid in the time granted for the revision.

Figure 1. The phenotype of $\Delta mdm31$ mutants is very different from that of ERMES deletion strains.

Upper panel: Radiolabeled Hsp60 protein was incubated with semi-intact cells of the indicated strains. Please note, that the deletion of Mdm31 basically abolished the mitochondrial protein import competence.

Lower panel: Deletion of Mdm31 leads to a strong growth defect on all carbon sources but – in contrast to the loss of ERMES – does not show synthetic defects with the loss of Tom70.

To explicitly mention the importance of contact sites for the mitochondrial lipid content, we now added the following sentence into the discussion: ‘However, changes in the mitochondrial lipid composition in the contact site mutants is expected to contribute to the reduced abundance of mitochondrial proteins, particularly of those that reside in the inner membrane.’

(2) The authors depleted only Mdm34 as a test to analyze the effect of the loss of ERMES on protein transport and said, "tethering of the two organelles as the artificial Tom70-GFP-Ubc6 tether construct, which bridges mitochondria and ER membranes, did not mitigate the import defect observed in the semi-intact cell assay". However, I am wondering why the authors did not test the loss of other ERMES subunits to rule out the possibility that the observed effect is specific to Mdm34, not the ERMES. Therefore, they should at least test the depletion of another ERMES subunit such as Mdm12. Mdm12 can be presumably depleted quickly by auxin-inducible degron.

We designed CRISPRi guide RNAs for MDM12 as suggested by the referee. Unfortunately, they did not deplete the MDM12 mRNA to more than 50% of the normal level (in the best case of the MDM12.3 guide RNA). We show this here for inspection by the referee as Figure 2.

Figure 2. CRISPRi-interference of MDM12 only moderately depletes the levels of MDM12 mRNA. A. Scheme for MDM12 knockdown. B. qPCR results of cells that were grown on glucose medium and treated for 2 h with ATc.

Therefore, in order to address your request, we did it the other way around and knocked down Tom70 in $\Delta mdm12$ cells. As shown in the novel Appendix Fig. S2E and F, this again strongly compromised the mitochondrial import of Oxa1 in the assay with semi-intact cells. Thus, ERMES (and not only Mdm34) serve as Tom70 partners in the ER-to-mitochondria transfer of ER-SURF substrates.

Specific points.

Figure 1B - Cellular localization of the accumulated Oxa1 precursor in the absence of Mdm34 should be shown.

Our mass spec data showed that Oxa1 fractionates with the ER in the Mdm34-depleted cells. In order to validate this, we now added microscopy images with GFP-tagged Oxa1 expressed in Mdm34-depleted cells. In particular, upon repression of Cdc48 (and thus ER-associated degradation), Oxa1-GFP signals were observed on the perinuclear ER. This is shown here for inspection by the referee. Since the accumulation of Oxa1 and other mitochondrial precursor proteins was described in depth recently in another study (Knöringer et al. 2023 MBoc 34, ar95), we decided not to show this experiment here.

GAL-CDC48
Oxa1-GFP
CRISPRi MDM34

For reference from Simakin et al., 2023:

Fig. 3. A fraction of Oxa1-GFP is present on the ER in Mdm34-depleted cells.

Top panels: Oxa1-GFP was expressed in cells in which Cdc48 was repressed by use of a regulatable GAL promoter. Upon knock-down of Mdm34, Oxa1-GFP showed the perinuclear staining that is characteristic for ER proteins. However, at these steady state conditions, the predominant fraction of Oxa1-GFP is present in mitochondria.

Bottom panels: Images showing the characteristic patterns of proteins residing in the ER (left) and mitochondria (right), respectively, for comparison. The images were taken from a recent publication from our lab Simakin, Koch et al., 2023.

Fig. 1D - The EM images did not convincingly indicate that intracellular organization or more specifically, organelle contacts, were not affected in semi-intact cells. The ER, mitochondria, and ER-mitochondria contacts should be analyzed by fluorescent microscopy.

We stained the ER and mitochondria in the different mutants and show these images in the novel Fig. EV4B as requested. These pictures show that the structure of mitochondria is strongly compromised once Mdm34 is depleted. This is consistent with the conclusions drawn on basis of the EM pictures. However, also by fluorescence microscopy, contact sites cannot be directly seen unless specific reporters are used for example by use split-GFP. The relevance of Mdm34/ERMES and Tom70 for contact site formation has been characterized and reported in the past in different studies (Elbaz-Alon et al. 2015 Cell Rep 12, 7-14; Murley et al, 2015. J Cell Biol 209,3 539-548).

Fig. 1H, I - The effects of Tether on the semi-intact cell import in the absence of Dj1 and Mdm34 are not clear, so they should be quantified. The gel may show that the import defects were mildly recovered by Tether?

As requested by the referee, we tested the effect of the tether construct in the import experiments with semi-intact cells of the $\Delta tom70$ Mdm34-depletion mutant. As shown in the novel Appendix Fig. 2A, expression of the ER-mito-tether did not rescue the import defect for Oxa1. This experiment was carried out three times independently and quantified as suggested. The quantification is shown as novel Appendix Fig. 2B.

Figure 2 - The time course of the decrease in the ER-mitochondria contact sites had better be shown in the Mdm34 knockdown experiment.

The experiment shows the time course of the depletion of the MDM34 mRNA (Fig. 2C) and protein (Fig. 2D). In addition, we show the time course of the changes of the mitochondrial network (Fig. 2E). We hope that this addresses the request of the referee.

Fig. 2 - On the basis of Fig. 2B, the authors said that they used Glu media for further analyses, but the plates for Fig. 2E appear to contain Gal (see page 26, line 2 from the bottom).

We changed the text and describe now in detail which carbon sources were used for the different experiments.

Figs. 3, 4, and 5 - Fig. 3 showed the data taken for Glu media, but Figs. 4 and 5 showed those taken for Gal media? The authors should explain the reason for using different media in these experiments.

Again, we explained the description of the carbon sources more explicitly. We agree that this was not always clear and in the initial version, we sometimes did not mention the carbon source when we thought that this is not relevant. However, now we added information to all the experiments shown.

Fig. 3F - The authors said that, in Mdm34 depleted cells, the amounts of matrix and IM proteins were affected while those of OM and IMS proteins were not affected. However, this could simply reflect the decreased membrane potential across the IM due to altered lipid profiles (pointed out above).

We therefore now tested the relevance of the membrane potential in the import assay with semi-intact cells. Of course, complete dissipation of the membrane potential abrogates protein import. However, mutants with reduced respiration, such as those lacking an active respiratory chain are still fully able to import proteins into mitochondria of semi-intact cells. We show this now in the novel Appendix Figure S2G, using $\Delta cox18$ cells as an example for a mutant that lacks cytochrome c oxidase and therefore has a reduced membrane potential (Souza et al. 2000. JBC 275, 14898-14902).

Fig. 4A and B - The possible changes in the lipid profiles in each organelle (pointed out above) could affect the organelle densities, thereby leading to contamination of mitochondrial proteins in other organelle fractions.

This is why we purified the ER and mitochondria via affinity tags. Thereby, the purification procedure does not rely on organelle densities which indeed will be changed if ER-mitochondria contact sites are absent. The results of the affinity purified organelles are shown in Figures 5A-E and EV5A-F.

Fig. 4C- Organelle shapes are not well seen in these EM pictures. The authors should check the organelle shapes by fluorescence microscopy.

We now added pictures of the cells from fluorescence microscopy as novel Fig. EV4B as requested.

Methods - The used media should be described in detail (only the carbon sources were described). Are they synthetic media or not? How much is the ATc concentration?

We now added information on the carbon sources and ATc concentrations throughout the text.

Referee #3:

The manuscript by Herrmann et al. describes that the mitochondria-ER contact sites mediated by ERMES and Tom70 are critically important for the localization of the mitochondrial proteins utilizing the ER-SURF pathway. The pathway has been recently identified as a way to mitochondria for a subset of mitochondrial inner membranes proteins, such as Oxa1, that involves the surface of the ER as a platform on the way to mitochondria.

This study identifies ERMES and Tom70 as two parallel routes, by which the ER-SURF substrates are transferred to mitochondria. Both, ERMES and Tom70 are involved in creation of independent contact sites

between the ER and mitochondria. Dysfunction of both results in a strong impairment of protein import, changes in membrane signatures and the accumulation of mitochondria precursor proteins on the surface of the ER.

Major points

- It is important to show direct interactions between mitochondrial precursors using ER-SURF with the components of ERMES or other contact sites specific components. Presence or absence of direct interactions will lead to different interpretations on the requirements of the contact sites and the proteins, which form them. Do they provide the space regulation or constitute a direct step of precursors passing from the ER to the mitochondrial translocases.

Inspired by the suggestion of this referee we now tested for a direct interaction of Mdm34 with ER-SURF substrates that are on transit from the ER to mitochondria. To this end, we again used the import assay with semi-intact yeast cells to which radiolabeled precursor proteins were added. We indeed found that the Oxa1 precursor (thus the cytosolic Oxa1 species) was efficiently co-immunoprecipitated with Mdm34-HA whereas the mature form (thus the intramitochondrial Oxa1 species) was not. We now show this interesting result as novel Fig. 6C. Basically the same result was also seen with another ER-SURF substrate that we tested (Cox5A(Oxa1)) and we show this additional data item as novel Appendix Fig. S2H.

We thank the referee for this interesting suggestion which indeed points at a direct role of ERMES in precursor transfer. This is further supported by our observation that tether constructs cannot take over the role of ERMES in precursor transfer (see also the results from the novel Appendix Fig. S2A, B).

- Some of the described effects should be repeated in the absence of Lam6 and Djp to exclude a contribution from the multiple involvements of Tom70 (beyond the ER-mito contact sites)

As suggested, we now repeated the import assays with semi-intact cells that lacked the following protein pairs: Tom70/Mdm34, Djp1/Mdm34, Lam6/Mdm34 and Tom70/Mdm12. These additional data are now shown as Appendix Figures S2C, D, E and F. The results show that the combined absence of Tom70 and ERMES leads to the most severe defect, but that the deletion of Djp1 or Lam6 in an ERMES null mutant induces a partial import defect. This is consistent with the observed growth phenotypes of the mutant (Fig. 3B) and the partially redundant function of Djp1 and Lam6 as ER-bound Tom70 interactors as shown in our model (Fig. 6F). We thank the referee for suggestion of this interesting experiment.

Minor comments

- The proteomic data should be analyzed and discussed from the perspective of a hypothetical specificity of two different contacts sites. Lack of specificity argues for the indirect architectural role of contacts sites proteins fulfilling the space requirement for the ER-SURF pathway

Inspired by this wonderful idea of the referee we now added the novel Appendix Fig. S1 which shows the Tom70 and ERMES-dependent clients in a Venn diagram.

- Fig 4C, S4A: the labeling in the figure legend is missing.

Corrected: we describe now what M, N and V stands for.

Dear Hannes,

Thank you for the submission of your revised manuscript to EMBO reports. We have now received the reports from the referees who were asked to assess the revised version. All reports are copied below my signature.

While both referees support publication, referee #2 suggested to perform the previously suggested experiments using Vps13(D716H). Please address this concern in a point-by-point response and ensure to prominently mention and discuss the possibility that altered lipid composition might contribute to the observed changes on protein import.

Browsing through the manuscript myself, I noticed a few editorial things that we need before we can proceed with the official acceptance of your study.

- Please reduce the number of keywords to 5.
- Please update the 'Conflict of interest' paragraph to our new 'Disclosure and competing interests statement'. For more information see <https://www.embopress.org/page/journal/14693178/authorguide#conflictsofinterest>
- Please remove the Author Contributions from the manuscript file and make sure that the author contributions in our online submission system are correct and up-to-date. The information you specified in the system will be automatically retrieved and typeset into the article. You can enter additional information in the free text box provided, if you wish.
- The reference Knöringer et al appears twice in the reference list, once as preprint and once as Mol Biol Cell article.
- Table EV1-EV4 should be renamed to Dataset EV1-EV4 with the corresponding callouts. The legends are correctly provided in a separate tab in each Excel file and should be removed from manuscript file.
- Appendix Table S1-S3 should be uploaded as Table EV1-EV3 with the corresponding callouts, and legends removed from the manuscript file and Appendix PDF, and included above the tables in the Excel files.
- The Appendix needs a table of content with page numbers. Please correct the callouts in the text to Appendix Figure S1-S2 (S is missing).
- "The Appendix PDF contains the following documents" should be removed from manuscript file.
- Our production/data editors have asked you to clarify several points in the figure legends (see below). Please incorporate these changes in the manuscript and return the revised file with tracked changes with your final manuscript submission.
 - 1) Please note that the legend for figure EV 3a is missing in the manuscript. This needs to be rectified.
 - 2) Please note that the legend for figures EV 3b-d is incorrectly labelled as 3a-c. This needs to be rectified.
 - 3) Please define the annotated p value * in the legend of figure 5f; as appropriate.
 - 4) Please indicate the statistical test used for data analysis in the legends of figures 3e; 5b-c, f; EV 5b-c.
 - 5) Please note that in figures 1g; 4d; 6b; there is a mismatch between the annotated p values in the figure legend and the annotated p values in the figure file that should be corrected.
 - 6) Please note that the box plots need to be defined in terms of minima, maxima, centre, bounds of box and whiskers, and percentile in the legends of figures 3f; 5f; EV 3d.
 - 7) Please note that information related to n is missing in the legend of figure 5f. Although 'n' is provided, please describe the nature of entity for 'n' in the legends of figures 3f; 4d; EV 3d.
 - 8) Please note that the error bar is not defined in the legend of figure 1g.
- As a general note: We recommend that the individual data from each experiment should be plotted if $n < 5$, alongside an error bar. It helps in visualizing the distribution of measurements.
- Finally, EMBO Reports papers are accompanied online by A) a short (1-2 sentences) summary of the findings and their significance, B) 2-3 bullet points highlighting key results and C) a synopsis image that is 550x300-600 pixels large (width x height) in PNG for JPG format. You can either show a model or key data in the synopsis image. Please note that the size is rather small and that text needs to be readable at the final size. Please send us this information along with the revised manuscript.

- On a different note, I would like to alert you that EMBO Press offers a new format for a video-synopsis of work published with us, which essentially is a short, author-generated film explaining the core findings in hand drawings, and, as we believe, can be very useful to increase visibility of the work. This has proven to offer a nice opportunity for exposure i.p. for the first author(s) of

the study. Please see the following link for representative examples and their integration into the article web page:
https://www.embopress.org/video_synopses
<https://www.embopress.org/doi/full/10.15252/emj.2019103932>

With kind regards,

Martina

Referee #2:

This is a reviewed version of the previously submitted manuscript to this journal. The authors responded to many of my concerns and added substantially new data to the manuscript, which strengthen the manuscript a lot. However, it is a bit disappointing that the authors could not do experiments with Vps13(D716H) to rule out the possibility of the indirect effects of lipid composition changes on protein import. I am wondering if the editor can give additional time to the authors to perform this experiment. Here are some minor points for correction.

Page 3 line10 -- The ATPase Spf1 (P5A ATPase in human) -> The P5A-ATPase Spf1 in yeast/ATP13A1 in human?

Page 28, line 4 -- Excitation of 510 nm and 575 nm were used for mNeogreen and mScarlet-I respectively, -- These excitation wavelengths are correct?

Legend to Fig S3 is missing.

Referee #3:

The authors adequately and satisfactorily addressed all the concerns.

Re: EMBO reports manuscript EMBOR-2023-58090V2
"The ER-SURF pathway uses ER-mitochondria contact sites for protein targeting to mitochondria"

Point-by-point response

Referee #2

1. This is a reviewed version of the previously submitted manuscript to this journal. The authors responded to many of my concerns and added substantially new data to the manuscript, which strengthen the manuscript a lot. However, it is a bit disappointing that the authors could not do experiments with Vps13(D716H) to rule out the possibility of the indirect effects of lipid composition changes on protein import. I am wondering if the editor can give additional time to the authors to perform this experiment. Here are some minor points for correction.

We are glad to see that the referee feels that our revision strengthened our study. We agree with the referee that the loss of the two ER-to-mitochondrial contact sites will alter the lipid composition of mitochondria and that therefore the changes in protein distribution that we see might be influenced by these changes. The suggested experiment with the Vps13(D716H) mutant is interesting. This mutant survives the loss of the two ER-mitochondria contact sites, however, even in this suppressor, the ER-to-mitochondria lipid flux will be altered, and the lipid composition of mitochondria will not be identical to that of wild type cells. Thus, even if data from this mutant were included, the limitation of a potential influence by altered lipids will remain. In order to address this valid point of the referee, we now clearly mentioned the influence of changed lipid compositions as a limitation of our study. We added statements to this aspect to the results and the discussion. However, we still want to emphasize that the data we show here, in particular those from the in vitro import experiments with semi-intact cells and the observed accumulation of mitochondrial proteins on affinity-purified ER membranes, clearly document a direct role of ER-mitochondria contact sites for the precursor targeting to mitochondria.

2. Page 3 line10 -- The ATPase Spf1 (P5A ATPase in human) -> The P5A-ATPase Spf1 in yeast/ATP13A1 in human?

We changed the text as suggested to 'P5A-ATPase (Spf1 in yeast, ATP13A1 in humans)'

3. Page 28, line 4 -- Excitation of 510 nm and 575 nm were used for mNeogreen and mScarlet-I respectively, -- These excitation wavelengths are correct? Legend to Fig S3 is missing.

We changed the text as suggested to 'Excitation of 510 nm and 575 nm were used for mNeogreen and mScarlet-I respectively'. We also added the legend to Figure S3A (now EV3A).

Points raised by the editor

1. While both referees support publication, referee #2 suggested to perform the previously suggested experiments using Vps13(D716H). Please address this concern in a point-by-point response and ensure to prominently mention and discuss the possibility that altered lipid composition might contribute to the observed changes on protein import.

We added the sentence you suggested to the results (final statement) and to the discussion. We hope that mentioning this limitation of the study addressed the concern adequately.

2. Please reduce the number of keywords to 5.

Done

3. Please update the 'Conflict of interest' paragraph to our new 'Disclosure and competing interests statement'. For more information see <https://www.embopress.org/page/journal/14693178/authorguide#conflictsofinterest>

There are still no competing interests, also not according to your new rules. We changed the title of the paragraph.

4. Please remove the Author Contributions from the manuscript file and make sure that the author contributions in our online submission system are correct and up-to-date. The information you specified in the system will be automatically retrieved and typeset into the article. You can enter additional information in the free text box provided, if you wish.

We removed the Author Contributions and made sure that the information in the online submission is correct

5. The reference Knöringer et al appears twice in the reference list, once as preprint and once as Mol Biol Cell article.

Corrected

6. Table EV1-EV4 should be renamed to Dataset EV1-EV4 with the corresponding callouts. The legends are correctly provided in a separate tab in each Excel file and should be removed from manuscript file.

Done

7. Appendix Table S1-S3 should be uploaded as Table EV1-EV3 with the corresponding callouts, and legends removed from the manuscript file and Appendix PDF, and included above the tables in the Excel files.

Done

8. The Appendix needs a table of content with page numbers. Please correct the callouts in the text to Appendix Figure S1-S2 (S is missing).

Done

9. "The Appendix PDF contains the following documents" should be removed from manuscript file.

Done

Points raised by the production/data editors

Our production/data editors have asked you to clarify several points in the figure legends (see below). Please incorporate these changes in the manuscript and return the revised file with tracked changes with your final manuscript submission.

We provide a manuscript with the changes highlighted attached to this response document.

1) Please note that the legend for figure EV 3a is missing in the manuscript. This needs to be rectified.

Done

2) Please note that the legend for figures EV 3b-d is incorrectly labelled as 3a-c. This needs to be rectified.

Done

3) Please define the annotated p value * in the legend of figure 5f; as appropriate.

Done

4) Please indicate the statistical test used for data analysis in the legends of figures 3e; 5b-c, f; EV 5b-c.

Done

5) Please note that in figures 1g; 4d; 6b; there is a mismatch between the annotated p values in the figure legend and the annotated p values in the figure file that should be corrected.

The figure legends provide information about how the p values were corrected and about how the asterisks have to be interpreted. The asterisks in the figure provide information about the actual data. We checked this again and figure and legend are correct.

6) Please note that the box plots need to be defined in terms of minima, maxima, centre, bounds of box and whiskers, and percentile in the legends of figures 3f; 5f; EV 3d.

Done

7) Please note that information related to n is missing in the legend of figure 5f.

Although 'n' is provided, please describe the nature of entity for 'n' in the legends of figures 3f; 4d; EV 3d.

Done

8) Please note that the error bar is not defined in the legend of figure 1g.

Done

9) As a general note: We recommend that the individual data from each experiment should be plotted if $n < 5$, alongside an error bar. It helps in visualizing the distribution of measurements.

We now added the data points of all measurements to the box plots in our study.

10) Finally, EMBO Reports papers are accompanied online by A) a short (1-2 sentences) summary of the findings and their significance, B) 2-3 bullet points highlighting key results and C) a synopsis image that is 550x300-600 pixels large (width x height) in PNG for JPG format. You can either show a model or key data in the synopsis image. Please note that the size is rather small and that text needs to be readable at the final size. Please send us this information along with the revised manuscript.

We added a text document with the texts for A) and B), and a jpg figure for C)

11) On a different note, I would like to alert you that EMBO Press offers a new format for a video-synopsis of work published with us, which essentially is a short, author-generated film explaining the core findings in hand drawings, and, as we believe, can be very useful to increase visibility of the work. This has proven to offer a nice opportunity for exposure i.p. for the first author(s) of the study. Please see the following link for representative examples and their integration into the article web page:

<https://www.embopress.org/doi/full/10.15252/emj.2019103932>

This is a great suggestion, and we will consider to produce such a movie once our study is finally accepted.

Dr. Johannes Herrmann
University of Kaiserslautern
Cell Biology
Erwin-Schroedinger-Strasse 13
Kaiserslautern D-67663
Germany

Dear Johannes,

Thank you for approving the final minor edits. I am very pleased to accept your manuscript for publication in the next available issue of EMBO reports. Thank you for your contribution to our journal.

Kind regards,

Martina
